# Performative Validity of Recourse Explanations

**Gunnar König**[1,2]**, Hidde Fokkema**[3]**, Timo Freiesleben**[4,5]**,**
**Celestine Mendler-Dünner**[1,6]**, Ulrike von Luxburg**[1,2]

[1]Tübingen AI Center, [2]University of Tübingen
[3]Korteweg-de Vries Institute for Mathematics, University of Amsterdam
[4]LMU Munich, [5]Munich Center for Machine Learning (MCML)
[6]ELLIS Institute Tübingen, Max Planck Institute for Intelligent Systems, Tübingen
`g.koenig.edu@pm.me`

## Abstract

When applicants get rejected by a high-stakes algorithmic decision system, recourse explanations provide actionable suggestions for applicants on how to change their input features to get a positive evaluation. A crucial yet overlooked phenomenon is that recourse explanations are *performative*: When many applicants act according to their recommendations, their collective behavior may shift the data distribution and, once the model is refitted, also the decision boundary. Consequently, the recourse algorithm may render its own recommendations *invalid*, such that applicants who make the effort of implementing their recommendations may be rejected again when they reapply. In this work, we formally characterize the conditions under which recourse explanations remain valid under their own performative effects. In particular, we prove that recourse actions may become invalid if they are influenced by or if they intervene on non-causal variables. Based on this analysis, we caution against the use of standard counterfactual explanation and causal recourse methods, and instead advocate for recourse methods that recommend actions exclusively on causal variables.

## 1 Introduction

Modern machine learning systems can significantly impact people's lives. Automated systems may determine whether someone receives a loan, is admitted to a graduate program, or gets invited to a job interview. In such high-stakes scenarios, applicants who receive an unfavorable decision – such as being rejected for a loan or a job interview – often seek guidance on how to improve their chances in the future. To address this need, recourse explanations are employed: they inform applicants of concrete changes they could implement to achieve the desired outcome from the system [Karimi et al., 2022]. For example, rejected loan applicants might be recommended to reduce their credit card utilization, and rejected job applicants might be advised to obtain a master's degree and reapply.

Ideally, recourse explanations are *valid* – meaning that if applicants follow the recommended actions (e.g., obtaining a master's degree), they will indeed receive the desired outcome when reevaluated by the machine learning model (e.g., be invited to the interview). In practice, however, deployed prediction models are rarely static. Model owners routinely monitor for distribution shifts and retrain their models to maintain performance. Recent work has shown that model updates can undermine the validity of recourse [Rawal et al., 2020, Upadhyay et al., 2021, Nguyen et al., 2023]. These studies examine how different types of distribution shift – such as data corrections, temporal drift, or geospatial variability – can affect recourse validity, and propose methods for generating recommendations that remain robust under such shifts.

39th Conference on Neural Information Processing Systems (NeurIPS 2025).

What these works overlook, however, is that by recommending certain actions, recourse itself can cause a distribution shift. We refer to this phenomenon as the *performativity of recourse explanations*. As a running example, consider a company using a machine learning model to screen candidates for interviews for a software engineering job (see Figure 1). Based on historical data, the model learns that applicants invited to interviews often hold a master's degree in software engineering or, alternatively, show regular activity on GitHub. Two recourse actions might therefore be recommended to rejected applicants: 1. Earn a master's degree in software engineering. 2. Increase GitHub activity by making regular commits. Both actions have performative effects: when implemented, they induce a distributional shift. However, they differ critically in their causal impact on qualification: While completing a master's degree imparts substantial knowledge in software engineering,

GitHub activity is merely a correlate of coding experience. Many tools exist that automatically generate daily commits, inflating an applicant's profile without improving their actual skills. If many applicants adopt such tools, the model will learn that GitHub activity no longer correlates with programming ability. As a result, future versions of the model may ignore the feature entirely—invalidating the very recourse that applicants were advised to pursue.

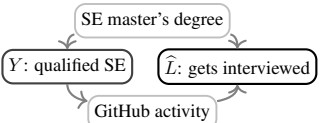

Figure 1: Simplified causal graph of features used to predict if applicants should be invited for a software engineering (SE) job interview.

In this work, we investigate whether recourse explanations remain valid under their own performative effects. Unrewarded recourse can impose significant burdens; thus, this question is particularly pressing in light of the social impact of recourse practices. When recourse fails, applicants may waste time and resources pursuing actions that ultimately do not improve their outcomes [Venkatasubramanian and Alfano, 2020].

**Contributions.** In Section 4, we introduce the two core concepts of our analysis. First, we formalize the *performative effect of recourse explanations* as causal interventions in response to recourse recommendations on acted-upon features in the data-generating process. Second, we introduce the notion of *performative validity*: a recourse explanation is *performatively valid* if, after applicants act upon it, they receive the desired outcome under the optimal post-recourse prediction model. In Section 5, we prove that recourse becomes performatively invalid if the implemented action carries predictive information about the post-recourse outcome. Building on this insight, we identify two mechanisms that can induce such invalidity: actions that are influenced by effect variables (Section 5.1), and actions that intervene on effect variables (Section 5.2). Then, in Section 6, we empirically study the performative validity of three recourse methods: counterfactual explanations [Wachter et al., 2017], causal recourse [Karimi et al., 2021], and improvement-focused causal recourse [König et al., 2023]. Our findings show that counterfactual explanations and causal recourse, by targeting non-causal variables, often result in performatively invalid recourse! In contrast, improvement-focused causal recourse remains valid across a wide range of data-generating processes.

## 2  Related Work

**Robust recourse:**  We refer to Karimi et al. [2022] for a comprehensive review on the literature of algorithmic recourse, and Mishra et al. [2021] for an overview on robust recourse. Most relevant for our work are studies that consider the validity of recourse when the underlying predictive model changes over time. Rawal et al. [2020] demonstrated empirically that recourse recommendations become invalid in the face of model shifts resulting from natural dataset shifts, including temporal changes, or geospatial variation. This has motivated the design of recourse that is robust to distributional shifts [Upadhyay et al., 2021, Nguyen et al., 2023, Jiang et al., 2024] and temporal factors [De Toni et al., 2024]. Similarly, Pawelczyk et al. [2020], Black et al. [2021], König et al. [2023] find that natural variations in model retraining–even when performed on the same data–can render previously recommended recourse invalid. In response, approaches have been designed that optimize for robustness to model multiplicity, hyperparameter choices, and stochasticity, among others [Dutta et al., 2022, Hamman et al., 2023, 2024]. We also focus on the validation of recourse under distributional shift. However, our approach is novel in examining performative distribution shifts—that is, endogenous shifts that are induced by the very provision of recourse itself.

**Performativity:** Predictive models may influence the very data they are later evaluated on—a phenomenon known as performativity [MacKenzie and Millo, 2003]. Implications for risk minimization have been studied under the umbrella of performative prediction [Perdomo et al., 2020a]. The framework formalizes two foundational concepts: *performative optimality* and *performative stability*. A model is performatively optimal if it is the best response to the data distribution induced by its predecessor. It is performatively stable if it remains optimal even after the distribution shift it itself induces. Building on these ideas, subsequent work has proposed algorithms that yield performatively optimal or stable models; We refer to Hardt and Mendler-Dünner [2025] for a comprehensive review of related work on performative prediction. Our work extends this line of research by developing a conceptual framework for performative recourse, analogous to performative prediction. Recourse that induces no change in $P(Y \mid X)$ corresponds to a natural analogue of a performatively stable point. We generalize this notion to recourse explanations that are rewarded even if the optimal model changes in response to the recourse-induced distribution shift.

**Strategic classification:** Strategic classification is a specific form of performativity in which agents deliberately manipulate their inputs to receive a favorable prediction from a model [Hardt et al., 2016]. A growing body of work investigates how to build models to guard against such strategic manipulations [Levanon and Rosenfeld, 2021, Chen et al., 2020, Zrnic et al., 2021, Chen et al., 2023, Haghtalab et al., 2021]. Thereby, a causal perspective has proven especially fruitful: In particular, the literature distinguishes between two types of user actions: gaming and improvement [Kleinberg and Raghavan, 2020, Miller et al., 2020, Ghalme et al., 2021, Tsirtsis et al., 2024, Efthymiou et al., 2025]. Gaming refers to actions on non-causal variables that change the model prediction $\hat{Y}$ without altering the true target $Y$, while improvement refers to actions on causal variables that affect $Y$ itself. While improvement affects only the marginal distribution $P(X)$, gaming may also alter the conditional distribution $P(Y \mid X)$ [Pfister et al., 2021, König et al., 2021, Horowitz and Rosenfeld, 2023]. We find that standard recourse methods—since they can inadvertently encourage users to game the system [König et al., 2023]—commonly induce shifts similar to those described in [Pfister et al., 2021, Horowitz and Rosenfeld, 2023]. By modelling the applicant's action as dependent on their own characteristics, we identify an additional source of performative invalidity: Recourse actions may leak otherwise unknown information about the applicant into the post-recourse distribution—even if recourse avoids gaming. As a consequence, further assumptions about the data-generating process are required to guarantee performative validity.

# 3 Preliminaries

We consider a supervised learning setting where the goal is to predict some target variable $Y$, such as an applicant's qualification as a software engineer. The prediction is based on a vector of observed features $X = (X_1, \ldots, X_p)$ in some space $\mathcal{X}$—for instance, university degrees or GitHub activity. Companies may only want to invite applicants above a certain qualification threshold. To formalize this, we binarize the target variable $Y$ using a threshold, formally $L := \mathbb{1}[Y \geq 0] \in \{0, 1\}$. We denote the conditional probability of a positive label given features as $h(x) = P(L = 1 \mid X = x)$, and the corresponding binary classifier as $\widehat{L}(x) = \mathbb{1}[h(x) \geq t_c]$, with decision threshold $t_c \in [0, 1]$.

**Causal model.** Algorithmic recourse concerns causal interventions that applicants perform in the world [Karimi et al., 2021, König et al., 2023]. To estimate these effects, a causal model is required. Throughout this work, we assume that the data-generating process can be modeled using an acyclic Structural Causal Model (SCM) $\mathbb{M} = \langle \mathbb{X}, \mathbb{U}, f \rangle$ [Pearl, 2009]: $\mathbb{X}$ includes all variables in the system, including the features $X_i$ and the target $Y$, $\mathbb{U}$ consists of independent noise variables $U_i$ capturing unobserved causal influences (for example, coding enthusiasm underlying GitHub activity), and $f$ is a set of structural equations that specify how each variable in $\mathbb{X}$ is generated from its causal parents and its corresponding noise. SCMs enable us to analyze the effects of interventions and counterfactual scenarios [Pearl, 2009]. For example, completing a master's degree can be modeled as an intervention. Formally, an intervention $a$ on a set of features $I_a$ is represented using Pearl's *do*-operator: $\mathrm{do}(a) = \mathrm{do}(\{X_i = \theta_i\}_{i \in I_a}) = \mathrm{do}(X_I = \theta_{I_a})$.

Each acyclic SCM induces a Directed Acyclic Graph (DAG) $\mathcal{G}$ that visualizes the causal relationships between the features. An example of such a graph can be found in Figure 1. For a variable $Z \in \mathbb{X}$, we denote its *direct causes* as $\mathrm{pa}(Z)$ (parents), its *direct effects* as $\mathrm{ch}(Z)$ (children), its full set of

direct and indirect causes as $\mathrm{an}(Z)$ (ancestors), its full set of direct and indirect effects as $\mathrm{de}(Z)$ (descendants). For example, in Figure 1, master's degree is a direct cause of qualification and thus a parent in the graph, and GitHub activity is a direct effect of qualification and hence a child in the graph. We abbreviate the direct effects of the target $Y$ as $E := \mathrm{ch}(Y)$, its causes as $C := \mathrm{an}(Y)$, and the direct parents of the direct effects as $S := \mathrm{pa}(X_E)$, also referred to as the spouses.

**Recourse methods.** Over the course of this paper, we will refer to three conceptually different recourse methods: Counterfactual Explanations (CE) [Wachter et al., 2017], Causal Recourse (CR) [Karimi et al., 2020b, 2021], and Improvement-focused Causal Recourse (ICR) [König et al., 2021]. Let $x$ be an applicant's current feature vector and $t_r \in [0, 1]$ the targeted success probability for the recourse recommendation. Further, let $c$ be a cost function that captures the effort required to implement the recommendation. The cost function takes two arguments: the applicant's observation $x$ and the suggested change, which, depending on the method, is either a causal intervention or a new datapoint $x'$. The methods are defined as follows:

**(CE)** A Counterfactual Explanations is defined as

$$a^{\mathrm{CE}}(x) = \arg\min_{x'} c(x, x') \quad s.t. \quad \widehat{L}(x') = 1.$$

CEs identify the closest input $x'$ to the current instance $x$ that flips the predicted outcome. They are widely used and many extensions have been proposed [Ustun et al., 2019, Karimi et al., 2020a, Dandl et al., 2020].

**(CR)** Causal Recourse is defined as

$$a^{\mathrm{CR}}(x) = \arg\min_{a} c(x, a) \quad s.t. \quad P(\widehat{L}(X^p) = 1 \mid x, \mathrm{do}(a)) \geq t_r,$$

where $X^p$ denotes the individual's observation after implementing the action $\mathrm{do}(a)$. Intuitively, CR identifies the least costly action that leads to a positive prediction with high probability. Unlike CEs, CR takes the causal relationships between features into account.

**(ICR)** Improvement-focused Causal Recourse is defined as

$$a^{\mathrm{ICR}}(x) = \arg\min_{a} c(x, a) \quad s.t. \quad P(L^p = 1 \mid x, \mathrm{do}(a)) \geq t_r,$$

where $L^p$ is the applicant's label after implementing the action. Intuitively, ICR recommends the least costly action that leads to a positive outcome with high probability. That is, while CE and CR directly focus on changing the prediction, ICR guides the applicant to become qualified. When the decision model is accurate, ICR reverts the decision with high probability, as shown in Proposition 1 of König et al. [2023].

The approaches differ fundamentally in the types of interventions they suggest: Where CE and CR may *suggest interventions on effects* that change the prediction without altering the applicant's qualification, ICR exclusively *intervenes on causes* and changes the qualification.

We note that for CR and ICR, two versions exist: The individualized version (**ind. CR** or **ind. ICR**) which is based on individualized causal effect estimates but requires knowledge of the SCM. And the subpopulation-based version (**sub. CR** or **ind. ICR**) that only requires the causal graph but resorts to less accurate average causal effect estimates (details in Appendix B). All theoretical results hold for both versions.

## 4 Performativity of Recourse Explanations

When rejected individuals implement recourse, they modify their characteristics and reapply. As such, recourse itself can shift the applicant distribution and, once the model is updated, also the decision boundary. We refer to this phenomenon as the *performativity of recourse*.

**Distribution Shift and Model Retraining.** To formalize the shift caused by recourse, we distinguish between the *pre-recourse* and *post-recourse* states of an applicant, denoting the post-recourse

state with the superscript $p$. Assuming that an $i.i.d.$ sample of rejected individuals implements recourse and reapplies, the new applicant distribution can be written as the mixture

$$P(L^m = l, X^m = x) := \alpha P(L = l, X = x) + (1 - \alpha)P(L^p = l, X^p = x \mid \widehat{L} = 0), \quad (1)$$

where superscript $m$ denotes mixture variables, $P(L, X)$ is the pre-recourse applicant distribution, $P(L^p, X^p \mid \widehat{L} = 0)$ the distribution of previously rejected applicants after implementing recourse, and $\alpha \in (0, 1)$ the mixing weight that determines the proportion of pre- versus post-recourse applicants. The optimal predictor for this updated applicant distribution becomes:

$$\widehat{L}^m(x) = \mathbb{1}[P(L^m = 1 | X^m = x) \geq t_c].$$

### 4.1 Performative validity

For individuals implementing recourse, it is fundamental that the validity of recourse is not affected by its performative effects. We refer to this requirement as *performative validity*.

**Definition 4.1 (Performative Validity).** Given a predictive model $\widehat{L}$, a recourse method is called *performatively valid* with respect to a set of individuals $\mathcal{X}' \subseteq \mathcal{X}$, if for all $x \in \mathcal{X}'$ and all $\alpha \in [0, 1]$ the optimal updated model $\widehat{L}^m$ satisfies $\widehat{L}^m(x) \geq \widehat{L}(x)$.

Intuitively, the definition requires that whenever recourse is effective for an individual $x \in \mathcal{X}'$ with respect to the original model, the updated model must also accept the applicant. Thereby, it should not matter how much data is collected, and thus the guarantee must hold irrespective of the mixing weight $\alpha$. Note that performative validity is related to the notion of *performative stability* [Perdomo et al., 2020b], which requires a model to remain optimal for the distribution it entails. As such, if the model is performatively stable w.r.t. the distribution map implied by recourse, it is also performatively valid. However, in general the converse does not hold.

### 4.2 A model of the post-recourse distribution

To obtain the post-recourse state of applicants, we assume that rejected individuals follow the actions recommended by the recourse algorithm while all other factors remain constant. To determine the causal effects of these actions on downstream variables, we assume access to the underlying structural causal model (SCM). In the SCM, each variable is a function of its observed and unobserved causes, formally $x_j := f_j(x_{\mathrm{pa}(j)}, u_j)$. Causal interventions $\mathrm{do}(a) = \mathrm{do}(X_{I_a} = \theta_{I_a})$ are implemented by replacing this function with a fixed assignment $x_j := \theta_j$.

Since recourse actions differ across applicants, we introduce a dedicated action variable $A$ and obtain the post-recourse state by replacing each intervened-upon structural equation $f_j$ with a function $f_j^p$ that switches between the normal assignment and the intervention based on $A$, that is

$$A := \begin{cases} a(x) & \text{if } \widehat{L}(x) = 0 \\ \mathrm{do}(\varnothing) & \text{otherwise} \end{cases}, \quad \text{and} \quad x_j^p := f_j^p(x_{\mathrm{pa}(j)}^p, u_j^p, a) = \begin{cases} \theta_{a,j} & \text{if } j \in I_a \\ f_j(x_{\mathrm{pa}(j)}^p, u_j^p) & \text{otherwise} \end{cases}.$$

The action $A$ causally depends on the pre-recourse observation $x$: first, via the decision model $\widehat{L}(x)$, which decides whether a recommendation is made, and second, via the recommendation $a(x)$. In short, we say that the action is influenced by $x$. To capture this influence, we include both the pre-recourse variables $X$ and post-recourse variables $X^p$ in the model. The causal relationships among pre-recourse variables are governed by the original structural equations $f$, while those among post-recourse variables follow the modified structural equations $f^p$.

Our model induces the causal graph in Figure 2(a). Notably, the pre- and post-recourse states are connected not only via $A$, but also via the unobserved causes $U$ and $U^p$. We typically assume that unobserved causes remain unchanged between the pre- and post-recourse states, though in some settings it is more realistic to allow them to change (we discuss this in Section 5.1).

## 5 Recourse May Invalidate Itself if It Relies on Effect Variables

After introducing the setup, we now formally characterize conditions under which performative validity holds. The following theorem shows that performative validity is guaranteed if the proposed action $A$ is conditionally independent of $L^p$ given $X^p$.

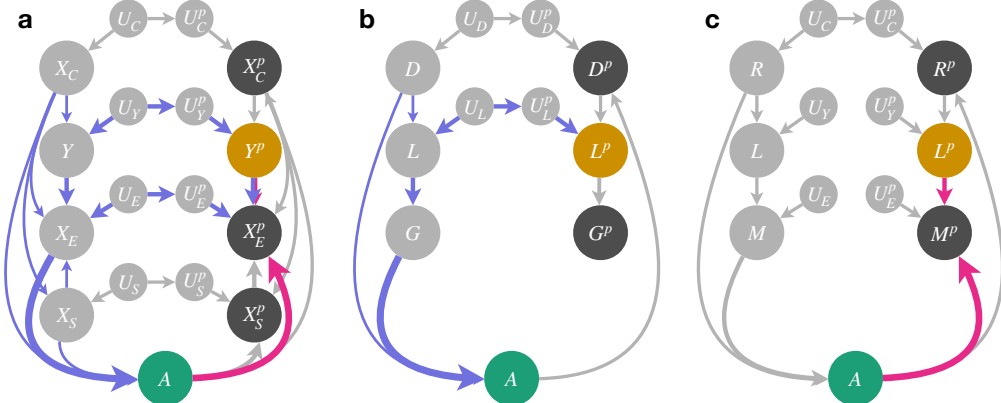

Figure 2: **Causal graphs.** In each graph, we color open paths between action $A$ and post-recourse target $Y^p$ given post-recourse observations $X^p$ via the incoming and outgoing edges, and highlight the decisive edges $X_E \to A$ and $A \to X_E^p$ using thicker arrows. **(a)**: We illustrate the relationships between the target $Y$ and its causes $X_C$, its direct effects $X_E$, and its spouses $X_S$, as well as their relationships with their post-recourse counterparts $Y^p, X^p$. In general, the pre- and post-recourse variables are connected both via the action $A$ and the unobserved causal influences $U, U^p$ (Section 4.2), and there are open paths via $X_E \to A$ and $A \to X_E^p$ (Theorem 5.2). **(b)**: The causal graph for Example 5.3, where $X_C = D$ indicates having a master's degree and $X_E = G$ indicates the GitHub activity. Since only interventions on causes are suggested, only the path via $G \to A$ is open. **(c)**: The causal graph for Example 5.8, where $X_C = R$ is the general risk score and $X_E = M$ indicates the mood. Since the unobserved causal influences are resampled, only the path via $A \to M^p$ is open.

**Proposition 5.1** (**Uninformative actions imply performative validity**). *Consider any recourse method and all points $x$ that would have been accepted by the original pre-recourse model, that is, $\widehat{L}(x)=1$. Then the optimal original and post-recourse models $h$ and $h^p$ are the same on all such points $x$ if and only if observing whether an intervention was performed $\mathbb{1}[A \neq \mathrm{do}(\varnothing)]$ does not help to predict post-recourse:*

$$\mathbb{1}[A \neq \mathrm{do}(\varnothing)] \perp\!\!\!\perp L^p \mid X^p \qquad \Leftrightarrow \qquad h(x) = h^p(x) \quad \forall x : \widehat{L}(x) = 1.$$

*As a result, we can guarantee performative validity if observing the intervention does not help to predict the post-recourse outcome, that is*

$$A \perp\!\!\!\perp L^p \mid X^p \qquad \Rightarrow \qquad \forall x : \widehat{L}^m(x) \geq \widehat{L}(x).$$

To evaluate whether $A$ is conditionally independent of $L^p$ given $X^p$, we consider the causal graph in Figure 2(a). According to standard results in causal inference (see Appendix A), two variables are conditionally independent in the data if they are d-separated in the graph. From the graph, we can identify two potential reasons why $A$ may not be d-separated from $L^p$ given $X^p$:

**Theorem 5.2** (**Two Sources of Invalidity**). *Consider the retrained post-recourse model $\widehat{L}^m$ of Eq. (1) and assume there are no unobserved confounders. Then there exist only two potential sources of performative invalidity:*

1. $X_{\mathrm{de}(Y)} \to A$, *that is, recourse actions $A$ are influenced by effect variables, meaning that the decision model $\widehat{L}(X)$ or recourse method $a(X)$ causally depends on effects $X_{\mathrm{de}(Y)}$.*
2. $A \to X_E^p$, *that is, recourse actions $A$ intervene on effect variables, meaning that the recourse method suggests interventions on direct effects of the prediction target.*

*Proof.* (Sketch) If $A$ is d-separated from $Y^p$ given $X^p$ in the graph, then $A \perp\!\!\!\perp Y^p | X^p$ in the distribution induced by the corresponding SCM, and performative validity can be guaranteed (Proposition 5.1). We observe that in the general graph $A$ is d-connected with $Y^p$ given $X^p$, but becomes d-separated if we remove two edges: (1) the incoming edge $X_{de(Y)} \to A$, which represents that the recommended action may be influenced by the pre-recourse effects, and (2) the outgoing edge $\to X_E^p$, which represents that the actions may intervene on the post-recourse direct effects. $\square$

Theorem 5.2 shows that performative validity can be guaranteed if we avoid using effect variables. Applied to our running example we can guarantee validity if we avoid using the effect variable GitHub activity, that is, (1) neither the decision model nor the recourse algorithm are influenced by information about GitHub activity to arrive at their outputs, and (2), recourse does not suggest to change the GitHub activity. Conversely, if we rely on the effect variable GitHub activity, it is unclear whether recourse is performatively valid. To better understand whether and under which assumptions relying on effects leads to performative invalidity, we now study each of the two sources in detail.

## 5.1 Performative invalidity due to actions that are influenced by effect variables

In this section, we find that actions that are influenced by effect variables may indeed cause performative invalidity. Intuitively, the reason is that when actions are influenced by effect variables, they may contain relevant information about the unobserved causal influences $U$ that otherwise could not be obtained from the post-recourse observation. When the actions are implemented, this information leaks into the post-recourse variables and their relationship with the target may change.

**Example 5.3** (**Interventions that are influenced by effects may cause performative invalidity**). Let $X_C \sim \text{Bin}(0.5)$ be whether someone has a degree, $L$ the applicant's qualification, $X_E$ whether someone has significant GitHub activity, and let $t_c = 0.5$ be the decision threshold. Let furthermore $U_L \sim \text{Unif}(0, 1)$ capture the applicant's autonomy, an unobserved cause of qualification. Furthermore, suppose that GitHub activity is only predictive of autonomy in the subpopulation of applicants without a degree.

$$L := \begin{cases} \mathbb{1}[U_L > 0.55] & X_C = 0 \\ \mathbb{1}[U_L > 0.45] & X_C = 1 \end{cases} \qquad X_E := \begin{cases} L & X_C = 0 \\ 1 & X_C = 1 \end{cases}$$

The causal graph is visualized in Figure 2 (b). In this setting, the action $A$ is influenced by the effect variable (GitHub activity $X_E$) and captures information about the unobserved cause (autonomy $U_Y$): Applicants are rejected if and only if they have no GitHub activity, which only happens if they have low autonomy $u_L \in [0, 0.55]$. When those low-autonomy applicants reapply after getting a degree, resulting in $X^p = (1, 1)$, the relationship between having a degree and qualification changes: Pre-recourse, it is unclear whether individuals with a degree $X = (1, 1)$ have high or low autonomy; Post-recourse it is more likely that applicants with a degree have low autonomy and are thus less qualified. This can indeed lead to invalidity: If the model is updated on a mixture with only 20 percent post-recourse applicants, the updated model would already reject everyone with observation $(1, 1)$. All details are reported in Appendix D.1.

**The result is concerning, since it proves that there are settings where performative validity cannot be guaranteed, even if only interventions on causes are suggested.** To avoid performative invalidity in such settings, the only remedy is to change the decision model such that it does not rely on effect variables (this follows from Theorem 5.2). However, the model may not be in the explanation provider's control, and abstaining from using effect variables may result in a significant drop in predictive accuracy.

As follows, we present two assumptions that allow us to partially recover from this sobering result. We first observe that in Example 5.3 the action could only leak information because the unobserved causal influence, the applicant's autonomy, remains the same pre- and post-recourse. This may not always be the case: Think of a scenario where a patient in a closed psychiatric facility applies to spend a day outside, and the decision is made based on the person's risk to harm themselves $Y$, which is predicted based on their average risk score $X_C$ and their score in a quick examination $X_E$ that captures the daily mood. Here, the unobserved causes could be the weather $U_Y$ and the selection of questions for the examination $U_E$ – factors that are resampled every day. More formally, the pre- and post-recourse unobserved causal influences $(U_Y, U_E)$ and $(U_Y^p, U_E^p)$ are independent (Assumption 5.4). When this assumption is met, no relevant information about the post-recourse target $Y^p$ can leak via the suggested action, and thus recourse methods that only intervene on causes can be guaranteed to be performatively valid (Proposition 5.5).

**Assumption 5.4** (**Unobserved Noise Changes Post-Recourse**). The unobserved causal influences of $Y$ and $X_E$ are resampled post-recourse, that is $U_Y$ and $U_Y^p$, as well as $U_E$ and $U_E^p$ are i.i.d..

**Proposition 5.5** (**Assumption 5.4 restores performative validity**). *If Assumption 5.4 holds, recourse methods that avoid interventions on effects are performatively valid.*

Second, we observe that the action in Example 5.3 can only change the post-recourse conditional distribution because it has access to information that otherwise cannot be obtained from the post-recourse observation. Specifically, we recall that in the example GitHub activity is only predictive of autonomy when the person has no degree, which is only the case pre-recourse. However, depending on the functional form of the causal relationships, the pre- and post-recourse observation may provide similar information about the unobserved causal influences. Specifically, we introduce Assumption 5.6 and show that it implies performative validity when the recourse actions do not intervene on effects (Theorem 5.7). Intuitively, the assumption ensures that any two observations that were generated with the same structural equations and unobserved influences provide the same information about $U$. Furthermore, it covers many naturally occurring types of structural equations, including linear additive and multiplicative noise models.

**Assumption 5.6** (**Marginalization with Invertible Aggregated Noise**). For an SCM with structural equations $f_E$ and $f_Y$, we assume that

(i) there exists an aggregation function $b_{\mathrm{agg}}$ and a structural equation $f_{\mathrm{agg}}$ such that we can rewrite the composition $f_E(f_Y(x_C, u_Y), x_S, u_E)$ as $f_{\mathrm{agg}}(x_C, x_S, b_{\mathrm{agg}}(u_Y, u_E))$, and
(ii) $f_{\mathrm{agg}}$ is invertible, that is, there exists $f_{\mathrm{agg}}^{-1}$ such that $f_{\mathrm{agg}}^{-1}(x_C, x_E, x_S) = b_{\mathrm{agg}}(u_E, u_Y)$.

**Theorem 5.7** (**Assumption 5.6 restores performative validity.**). *Let Assumption 5.6 be satisfied. Further assume that the unobserved causal influences stay the same post-recourse, that is $(U_Y, U_E) = (U_Y^p, U_E^p)$. Then, recourse methods that abstain from interventions on effects are performatively valid.*

## 5.2 Performative invalidity due to actions that intervene on effect variables

To show that interventions on effects may indeed cause performative invalidity, we show that invalidity may occur in a setting where we can rule out that actions that are influenced by effects are responsible. This is tricky, since we cannot enforce that recourse actions are not influenced by effects but intervene on effects at the same time: To ensure that interventions are not influenced by effects the decision model must be invariant to changes in the effect variables—and as a result, no actions that intervene on effects would be suggested anyway. Instead, to show that interventions on effects are responsible for performative invalidity, we focus on a setting where actions that are influenced by effects are not problematic. In Section 5.1 we introduced two such settings, captured in Assumptions 5.4 and 5.6. As follows, consider a formal version of the psychiatric facility setting where the unobserved causal influences are resampled post-recourse and thus Assumption 5.4 is met.

**Example 5.8** (**Interventions on effects cause performative invalidity**). Let $Y$ capture a patient's daily risk of harming themselves, $X_C$ the patient's general risk score, and $X_E$ a quick examination that aims to capture the daily mood. More formally, let $X_C \sim \mathcal{N}(0, 1)$, $U_Y \sim \mathcal{N}(0, 1)$, $U_E \sim \mathcal{N}(0, 1)$ and their causal relationships be governed by the following structural equations:

$$Y := X_C + U_Y, \qquad X_E := Y + U_E, \qquad L = \mathbb{1}[Y \geq 0]. \qquad (2)$$

The causal graph is visualized in Figure 2 (c). We suppose that recourse suggests intervening on $X_E$, that is, to give more favorable answers in the quick examination. Since giving more favorable answers per se does not improve the patient's daily mood, the action only makes the patient's mood *appear* favorable. When many applicants start to game the examination, the variable becomes less predictive of patient risk, and updated models will reduce their reliance on the feature. Formally, recourse changes $x_E$ to move rejected individuals onto the decision boundary $x_C = -x_E$. The updated decision boundary moves towards $x_C = 0$, such that recourse-implementing applicants with $x_C < 0$ get rejected when they reapply (confer Appendix E.1 for details). Since Assumption 5.6 holds too, the phenomenon occurs irrespective of whether the noise is resampled (Assumption 5.4).

The example demonstrates that interventions on effects may indeed cause performative invalidity. More generally, we can prove that recourse via interventions on effects always leads to a shift in the conditional distribution (Appendix E). Whether the shift is severe enough to lead to performative invalidity depends on the concrete setting. These results are concerning, since the two most widely used recourse methods, namely CE and CR, may suggest intervening on effects.

**Corollary 5.9.** *In a setting where the first source of invalidity (influence on $A$) can be excluded, that is if either Assumption 5.4 or 5.6 holds, recourse algorithms that abstain from interventions on direct effects are performatively valid but other recourse algorithms may not. Without additional assumptions, **performative validity can only be guaranteed for ICR, but not for CR and CE**.*

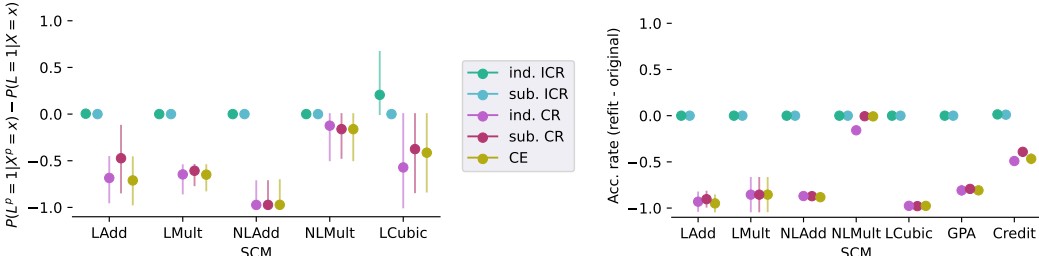

Figure 3: **Experimental results.** *(Q1, left):* The pointwise differences between pre- and post-recourse conditional distribution aggregated using the mean, the lines indicate the range. All values are averages over 10 runs. *(Q2, right):* The difference in acceptance rate (refit minus original), average (•) and standard deviation (lines) over 10 runs. While CE and CR lead to unfavorable shifts and performative invalidity, ICR is performatively valid in all settings.

# 6 Experiments Confirm That Interventions on Effects Must Be Avoided

Our theory suggests that the performative validity of recourse critically depends on the recourse method and the functional form of the underlying causal relationships. To investigate this empirically, we study the performative effects of CE, and the different versions of CR and ICR in synthetic and real-world settings. Specifically, we study the following classes of structural equations:

   (i) Additive noise (*LAdd*): $f_j(x, u) = l(x) + u$ where $l$ is linear
  (ii) Multiplicative noise (*LMult*): $f_j(x, u) = m(x)u$ where $m$ is linear in each dimension
 (iii) Nonlinear relations and additive noise (*NLAdd*): $f_j(x, u) = g(x) + u$ where $g$ nonlinear
  (iv) Nonlinear relations and multiplicative noise (*NLMult*): $f_j(x, u) = g(x)u$ where $g$ nonlinear
   (v) Polynomial noise (*LCubic*): $f_j(x, u) = (l(x) + u)^3$

We note that the first two settings (*LAdd* and *LMult*) satisfy Assumption 5.6, but the remaining settings do not. To enable pointwise comparisons of the conditional distributions, we rely on discrete noise distributions with finite support.

In addition, we include two real-world settings: College admission (*GPA*) and credit scoring (*Credit*). For *GPA* we assume the causal graph in [Harris et al., 2022] and fit a linear SCM with additive Gaussian noise on the dataset [OpenIntro, 2020]; For *Credit* we rely on the graph by Chen et al. [2023] and fit a random-forest based, nonlinear SCM with additive Gaussian noise [Yeh, 2009]. The *Cred* setting has eleven features; all others include one cause and one effect variable. To highlight the differences between methods, we choose the costs such that interventions on effects are more lucrative. Consequently, CE and CR intervene on the effect, while ICR only intervenes on the cause. To allow both sources of invalidity to come into effect, we always model the noise to stay the same post-recourse. We provide a detailed description of setup and results in Appendix F.

**Conditional distributions (Q1):** *Does the conditional distribution that is modeled by the predictor change? More formally, are $P(L^p = 1|X^p = x)$ and $P(L = 1|X = x)$ the same?*

For the synthetic settings with discrete support we compare the pre- and post-recourse conditional distributions by computing the pointwise differences $P(Y^p = 1|X^p = x') - P(Y = 1|X = x')$ and aggregating them using the minimum, the maximum, and the mean. When computing the mean, we weight the different points $x'$ according to their post-recourse probability in the population of rejected applicants, formally $P(X^p = x' \mid \widehat{L} = 0)$. The results are reported in Figure 3 (left). We find that methods that intervene on effects (CE and CR) significantly *decrease* the conditional probability of a favorable outcome across all settings. For example, in the nonlinear additive setting, the conditional probability of a favorable outcome decreases by between 70% and 100%. In contrast, for ICR, the conditional probability *remains the same in nearly every setting*, including but not limited to the two settings covered by Assumption 5.6. The exception is the setting *LinCubic*, where for ind. ICR the conditional probability of a favorable outcome *increases* by between 0% and 60%. This is consistent with our theory.

**Acceptance rates for the updated model (Q2):** *Does performative validity hold? Which percentage of the recourse-implementing applicants gets accepted by the original vs the updated models?*

To compare the pre- and post-refit acceptance rates, we compute the percentage of the resource-implementing applicants who get accepted by the original model vs an updated model that is trained on a mixture with $\alpha = 1/3$ post-recourse data. The results are reported in Figure 3 (right). We observe that the interventions on effects recommended by CE and CR not only change the conditional distribution, but also lead to a dramatic drop in the post-recourse acceptance rates. For example, in the GPA setting, the acceptance rate drops by nearly $80\%$ for CE and CR. In contrast, the acceptance rates for ICR remain unchanged across all settings, extending beyond our theoretical results.

In conclusion, the empirical results *confirm our theory*: Even recourse methods that only intervene on causes (ICR) may lead to a shift when they are influenced by effects but only methods that suggest to intervene on effects (CE and CR) can cause a shift in settings covered by Assumption 5.6. Extending *beyond our theoretical analysis*, we observe that ICR seldom leads to a shift, and when it does, the shift is not problematic. In contrast, CE and CR shift the distribution across all settings and cause severe performative invalidity. Our findings consistently confirm that **interventions on effects must be avoided**.

# 7  Discussion and Future Work

In this paper, we show that improvement-focused causal recourse (ICR), unlike counterfactual explanations (CEs) and causal recourse (CR), can maintain performative validity across a broad range of data-generating processes, including those involving resampling or additive noise. Empirically, ICR appears to remain valid even under more general conditions. A key direction for future research is to formally characterize the full class of data-generating processes under which ICR guarantees performative validity. Additionally, because full causal knowledge is rarely available in practice, it would be interesting to extend our framework to settings with incomplete causal knowledge, including cases that violate causal sufficiency.

This paper has examined the performative effects of recourse explanations from the perspective of the applicant. A promising direction for future work is to complement this view with the perspective of the model authority [Fokkema et al., 2024]. Model authorities may use recourse explanations to strategically steer applicants toward regions of the input space where model performance improves or institutional goals are better met. In this setting, our notion of recourse validity could be extended to actions that are valid post-recourse but not pre-recourse, offering a conceptual analogue to performative optimality [Perdomo et al., 2020a]. However, such steering raises important ethical concerns—particularly when it overlooks or conflicts with the applicant's own goals and values [Kim and Perdomo, 2022, Hardt et al., 2022, Zezulka and Genin, 2024].

We focused on single-step recourse – where applicants act on a one-time recommendation and then face the updated model. However, many real-world scenarios involve repeated decision-making, such as applicants reapplying for loans multiple times [Verma et al., 2022, Fonseca et al., 2023]. Building on insights from the performative prediction literature future research could investigate whether different recourse strategies converge to stable equilibria, and if so, analyze their properties and desirability.

**Practical Insight.** Counterfactual explanations are widely used in practice to guide applicants toward their desired outcomes. However, our findings show that, due to the performative effects of their actions, applicants may still fail to achieve these outcomes – even after following the recommended recourse. To prevent applicants from taking costly yet ineffective actions, model authorities should only give recourse recommendations that improve the qualification of the applicant. In particular, we advise against relying on standard counterfactual or causal recourse explanations. Thereby, we strengthen the position taken in previous work [König et al., 2023], which demonstrates the benefits of improvement-focused recourse from the perspective of the model authority.

## Acknowledgments and Disclosure of Funding

The authors are grateful to Anna Vollweiter for running preliminary, yet different, experiments in the early stages of the project. This work has been supported by the German Research Foundation through the Cluster of Excellence "Machine Learning - New Perspectives for Science" (EXC 2064/1 number 390727645) and the Carl Zeiss Foundation through the CZS Center for AI and Law. Freiesleben was additionally supported by the Carl Zeiss Foundation through the project "Certification and Foundations of Safe Machine Learning Systems in Healthcare".

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

## A  Causality Preliminaries

Causal graphs cannot only be used to visualize causal relationships but, under certain assumptions, also to reason about conditional independencies in the data. To do so, we first introduce the so-called $d$-separation criterion.

**Definition A.1** ($d$-separation). Two variable sets $X, Y$ are *d-separated* given a variable set $Z$, denoted as $X \perp_{\mathcal{G}} Y \mid Z$, if and only if, for every path $p$ from $X$ to $Y$ one of the following holds

(i) $p$ contains a chain $i \to m \to j$ or a fork $i \leftarrow m \to j$ where $m \in Z$.
(ii) $p$ contains a collider $i \to m \leftarrow j$ such that $m$ and all of its descendants are not in $Z$.

Under certain assumptions, $d$-separation in the graph can be linked to conditional (in)dependencies in the data: First, if the so-called Markov property holds, $d$-separation in the graph to implies independence in the data. Second, if faithfulness holds, independence in the data implies $d$-separation in the graph.

**Definition A.2** (Markov Property). Given a DAG $\mathcal{G}$ and a joint distribution $P$ over the nodes, this distribution is said to satisfy the *Markov property* with respect to the DAG $\mathcal{G}$ if

$$X \perp_{\mathcal{G}} Y \mid Z \implies X \perp\!\!\!\perp Y \mid Z.$$

for all disjoint vertex sets $X, Y, Z$

**Definition A.3** (Faithfulness). Given a DAG $\mathcal{G}$ and a joint distribution $P$ over the nodes, this distribution is said to be *faitful* to the DAG $\mathcal{G}$ if

$$X \perp\!\!\!\perp Y \mid Z \implies X \perp_{\mathcal{G}} Y \mid Z.$$

Throughout the paper, we only use the graph to read off conditional independence relationships. For instance, in Theorem 5.2 we only say that there *may* be a dependence via the open paths but make no claim that it must be present in the data. As such, we only need the Markov property for our results. The Markov property is met if the data is induced by SCMs with independent noise terms, or more generally, if there are no unobserved confounders.

**Proposition A.4.** *Assume that $P$ is induced by an SCM with graph $\mathcal{G}$. Then, $P$ is Markovian with respect to $\mathcal{G}$.*

*Proof.* See for example Proposition 6.31 in [Peters et al., 2017]. $\qquad\square$

## B  Individualized and Subpopulation-based Causal Recourse Methods

Karimi et al. [2020b] propose two versions of Causal Recourse (CR): an *individualized* and *subpopulation-based* version. The individualized version leverages counterfactuals to make causal effect estimates, which take the information about the individual obtained from the observed features $X$ into account. Therefore, so-called rung 3 causal knowledge is required, which is, for example, given by an SCM [Pearl, 2009].

However, the SCM is rarely readily available in practice, and thus the subpopulation-based version was introduced as a fallback that only requires access to the causal graph. The causal graph is easier to obtain, since it only captures whether nodes are causally related, but does not detail the functional form of their relationship. However, causal graphs do not enable the estimation of individualized treatment effects (rung 3 on Pearl's ladder of causation [Pearl, 2009]). Instead, causal graphs can be used to compute conditional average treatment effects, that is, interventional distributions for a subpopulation or similar individuals (rung 2). Specifically, causal graph-based treatment effect estimates can account for characteristics that are assumed to be unaffected by the intervention. In the graph, these are the non-descendants of the intervened-upon variables $X_{\mathrm{nd}(I_a)}$.

In the background sections, we introduced the individualized versions of the causal recourse methods. The definitions of the subpopulation-based versions of CR and ICR only differ in the conditioning set (highlighted red):

$$a^{\mathrm{CR}}(x) = \arg\min_{a} c(x, a) \quad s.t. \quad P(\widehat{L}(X^p) = 1 \mid x_{\mathrm{nd}(I_a)}, \mathrm{do}(a)) \geq t_r,$$

and

$$a^{\mathrm{ICR}}(x) = \arg\min_a c(x, a) \quad s.t. \quad P(L^p = 1 \mid x_{\mathrm{nd}(I_a)}, \mathrm{do}(a)) \geq t_r.$$

# C   Proofs of Section 5

Let us make two general remarks, before we start with the proofs. First, when the context allows, we shorten the notation of conditional distributions. Specifically, we sometimes drop the random variable and only write the corresponding observation. For example,

$$P(L^p = 1 \mid X^p = x^p, U_C = u_C) = P(L^p = 1 \mid x^p, u_C).$$

Second, some of our results prove that the conditional distribution of the underlying potentially continuous target $Y$ remains the same. These results imply performative validity.

**Lemma C.1.** *If $P(Y^p \leq y \mid X^p = x) = P(Y \leq y \mid X = x)$ for all $(x, y) \in \mathcal{X} \times \mathcal{Y}$, then $P(L^p = 1 \mid X^p = x) = P(L = 1 \mid X = x)$ for all $x \in \mathcal{X}$.*

*Proof.* The following string of equalities proves the result,

$$P(L^p = 1 \mid X^p = x) = P(Y^p \leq 0 \mid X^p = x) = P(Y \leq 0 \mid X = x) = P(L = 1 \mid X = x).$$

$\square$

As follows, we repeat and prove the results from Section 5.

**Proposition 5.1** (**Uninformative actions imply performative validity**). *Consider any recourse method and all points $x$ that would have been accepted by the original pre-recourse model, that is, $\widehat{L}(x){=}1$. Then the optimal original and post-recourse models $h$ and $h^p$ are the same on all such points $x$ if and only if observing whether an intervention was performed $\mathbb{1}[A \neq \mathrm{do}(\varnothing)]$ does not help to predict post-recourse:*

$$\mathbb{1}[A \neq \mathrm{do}(\varnothing)] \perp\!\!\!\perp L^p \mid X^p \qquad \Leftrightarrow \qquad h(x) = h^p(x) \quad \forall x : \widehat{L}(x) = 1.$$

*As a result, we can guarantee performative validity if observing the intervention does not help to predict the post-recourse outcome, that is*

$$A \perp\!\!\!\perp L^p \mid X^p \qquad \Rightarrow \qquad \forall x : \widehat{L}^m(x) \geq \widehat{L}(x).$$

*Proof.* In the first two steps, we prove the equivalence between the conditional independence and the stability of the predictor. In the last step, we use the equivalence to show that conditional independence of the action implies performative validity.

**Step 1** ($L^p \perp\!\!\!\perp \mathbb{1}[A = \mathrm{do}(\varnothing)] \mid X^p \Leftrightarrow P(L^p \mid X^p) = P(L^p \mid X^p, A = \mathrm{do}(\varnothing))$)**.** The first step holds by the definition of conditional independence.

$$\begin{aligned}
P(L^p \mid X^p, A = \mathrm{do}(\varnothing)) &= \frac{P(L^p, A = \mathrm{do}(\varnothing) \mid X^p)}{P(A = \mathrm{do}(\varnothing) \mid X^p)} \qquad \text{(Bayes' rule)} \\
&= \frac{P(L^p \mid X^p)P(A = \mathrm{do}(\varnothing) \mid X^p)}{P(A = \mathrm{do}(\varnothing) \mid X^p)} \quad (L^p \perp\!\!\!\perp \mathbb{1}[A = \mathrm{do}(\varnothing)] \mid X^p) \\
&= P(L^p \mid X^p).
\end{aligned}$$

**Step 2** ($P(L^p \mid X^p = x, A = \mathrm{do}(\varnothing)) = P(L \mid X = x)$)**.**

In the subpopulation of individuals where no intervention is performed the structural equations are the same pre- and post-recourse. Since we assume that all other factors remain the same as well, the pre- and post-recourse distribution are the same. More formally,

$$P(L^p \mid X^p, A = \mathrm{do}(\varnothing)) = P(L \mid X, A = \mathrm{do}(\varnothing)).$$

Furthermore, the action $A$ is a function of $X$ and as a result

$$P(L \mid X, A = \mathrm{do}(\varnothing)) = P(L \mid X).$$

**Note:** Only accepted individuals $\widehat{L}(x) = 1$ are recommended to change nothing $A = \mathrm{do}(\varnothing)$, and thus $P(Y|X, A = \mathrm{do}(\varnothing))$ is only defined for accepted $x$. As a result, step 1 and 2 only apply to $x$ where $\widehat{L}(x) = 1$, and our result is restricted to $x$ where $\widehat{L}(x) = 1$.

We recall that $h(x) := P(L = 1|X = x)$ and $h^p(x) := P(L^p = 1|X^p = x)$. So taken together, the conditional independence is equalivalent to equal pre- and post-recourse model in the acceptance region.

**Step 3** ($A \perp\!\!\!\perp L^p \mid X^p \Rightarrow \widehat{L}^m(x) \geq \widehat{L}(x)$)**.**

To obtain the final implication, we observe that when $h(x) = h^p(x)$ for all $x$ where $\widehat{L}(x) = 1$, it also holds that $\widehat{L}^m(x) \geq \widehat{L}(x)$: Any $x$ that the original classifier $\widehat{L}$ is also accepted by the updated classifier $\widehat{L}^m$, and for any $x$ that is rejected by $\widehat{L}$ the classification can only improve. Overall we get

$$A \perp\!\!\!\perp L^p \mid X^p$$
$$\Rightarrow \quad \mathbb{1}[A = \mathrm{do}(\varnothing)] \perp\!\!\!\perp L^p \mid X^p$$
$$\Rightarrow \quad h(x) = h^p(x) \text{ for all } x \in \mathcal{X} \text{ where } \widehat{L}(x) = 1$$
$$\Rightarrow \quad h(x) = h^m(x) \text{ for all } x \in \mathcal{X} \text{ where } \widehat{L}(x) = 1$$
$$\Rightarrow \quad \widehat{L}^m(x) \geq \widehat{L}(x).$$

$\square$

**Theorem 5.2** (**Two Sources of Invalidity**). *Consider the retrained post-recourse model $\widehat{L}^m$ of Eq.* (1) *and assume there are no unobserved confounders. Then there exist only two potential sources of performative invalidity:*

1. *$X_{\mathrm{de}(Y)} \to A$, that is, recourse actions $A$ are influenced by effect variables, meaning that the decision model $\widehat{L}(X)$ or recourse method $a(X)$ causally depends on effects $X_{\mathrm{de}(Y)}$.*
2. *$A \to X_E^p$, that is, recourse actions $A$ intervene on effect variables, meaning that the recourse method suggests interventions on direct effects of the prediction target.*

*Proof.* If there are no unobserved confounders, which is for example the case if the data generating mechanism can be written as a SCM with independent noise terms, then the Markov property holds, and $d$-separation in the causal graph implies independence in the data. If $A$ were to be $d$-separated from $Y^p \mid X^p$, it would thus hold that $A \perp\!\!\!\perp Y^p \mid X^p$, and as a consequence of Proposition 5.1 recourse would be performatively valid. However, if there are open paths between $A$ and $Y^p$ given $X^p$, then $A$ may be dependent on $Y^p$ given $X^p$ and recourse may be performatively invalid.

Given $X^p$, there are only a few possible open paths between $A$ and $X^p$. To follow the proof, we recommend inspecting Figure 2 first.

An arrow with asterisks, $*\!\!-\!\!*$ , indicates that the arrow could go both ways. We observe that the following paths are guaranteed to be closed:

- All paths via causes of $X_C^p$ are blocked: All causes $X_C^p$ are observed, and as a result any path of the form $Y^p \leftarrow X_i^p \leftarrow \ldots *\!\!-\!\!* A$ or $Y^p \leftarrow X_i^p \to \ldots *\!\!-\!\!* A$ is closed for any $i \in C$.

- All paths via effects of effects are blocked: All effects $X_E^p$ are observed, and as a result any path of the form $Y^p \to X_i^p \to \ldots *\!\!-\!\!* A$ is closed for any $i \in E$.

- All paths via spouses are blocked: Spouses are observed and thus any path $Y^p \to X_i^p \leftarrow X_j^p \to \ldots *\!\!-\!\!* A$ or $Y^p \to X_i^p \leftarrow X_j^p \leftarrow \ldots *\!\!-\!\!* A$ is closed for any $i \in E$ and $j \in S$.

All remaining paths either include the segment $A *\!\!-\!\!* \ldots *\!\!-\!\!* U_Y^p \to Y^p$ or $A \to X_j^p \leftarrow Y^p$ with $j \in E$. Whenever the action $A$ causally influences an effect variable the action is $d$-connected with $Y^p$, since the edge $A \to X_j^p \leftarrow Y^p$ is open whenever $X_j^p$ is observed. Paths including $A *\!\!-\!\!* \ldots *\!\!-\!\!* U_Y^p \to Y^p$ are only open if the effect variables $X_E$ causally influence $A$:

- Any path that involves a collider structure $A \ast\!\!-\!\!\ast \cdots \to Y \leftarrow U_Y$ or $A \ast\!\!-\!\!\ast \cdots \to X_j \leftarrow \cdots \leftarrow Y \leftarrow U_Y$ is closed if $X_{de(Y)} \not\to A$.

- All other paths include an edge $A \leftarrow X_i$ where $i \in \mathrm{de}(Y)$.

To summarize: Unless there is an edge $A \to X_E^p$ or $X_{\mathrm{de}(Y)} \to A$, $A$ is $d$-separated from $Y^p$ given $X^p$, and thus $A \perp\!\!\!\perp Y^p \mid X^p$, and as a consequence recourse is performatively valid. $\qquad \square$

# D    Proofs of Section 5.1

## D.1    Details of Example 5.3

**Interventions that are influenced by effects may cause performative invalidity**    We start by deriving the conditional probabilities $P(L \mid X)$. The variable $X = (X_C, X_E)$ can take three different values: $(0,0)$, $(0,1)$ or $(1,1)$. In the first two cases, $X_E$ perfectly predicts the label $L$, that is

$$P(L = 1 \mid X = (0,0)) = 0$$
$$P(L = 1 \mid X = (0,1)) = 1.$$

In the third case, $X_C = 1$. When $X_C = 1$, the value of $X_E$ is determined by $X_C$ (by definition of the SCM). Thus, the conditional probability reduces to

$$P(L = 1 \mid X_C = 1, X_E = 1) = P(L = 1 \mid X_C = 1)$$
$$= 0.55. \qquad \text{(by definition)}$$

Assuming that the predictor is Bayes optimal, that is $\widehat{L}(x) = \mathbb{1}[P(L = 1 \mid X = x) \geq 0.5]$, only applicants with observation $X = (0,0)$ are rejected. To move these rejected individuals across the decision boundary, the only possible action is to intervene on $X_C$ by setting $\mathrm{do}(X_C = 1)$, yielding a post-recourse observation $x^p = (1,1)$.

To derive the updated model, we need to determine the post-recourse probability of a favorable label for people with $x^p = (1,1)$. Therefore, we first derive the unobserved noise distribution for the individuals who implement recourse. By definition of the SCM, the unobserved noise for the rejected individuals is distributed according to

$$P(U_L \mid \widehat{L} = 0) = P(U_L \mid X = (0,0)) \sim \mathrm{Unif}(0, 0.55).$$

Thus we get that $P(L^p = 1 \mid X^p, \widehat{L} = 0) = P(L^p = 1 \mid \widehat{L} = 0) = P(U_L > 0.45) = 0.1/0.55$.

To obtain the prediction by the updated model $\widehat{L}^m$, we derive the mixture $P^m$ with $\alpha = 0.2$ and observe that the observation $X = (1,1)$ now gets a conditional probability of

$$P(L^m = 1 \mid X^m = (1,1)) = 0.2 * \frac{0.1}{0.55} + 0.8 * 0.55 = 0.4844 \ldots < t_c.$$

As such, the recourse outcome $(1,1)$ is not valid for (rejected by) the updated model. $\qquad \square$

## D.2    Independent noise assumption and proofs

**Proposition 5.5** (**Assumption 5.4 restores performative validity**). *If Assumption 5.4 holds, recourse methods that avoid interventions on effects are performatively valid.*

*Proof.* Note that there are no connections between $U$ and $U^p$ in the graph depicted in Figure 2(a), whenever $U^p$ is independently resampled. As the probability of intervening on effects is 0, the arrow from $A$ to $X_E^p$ is not present and we observe the $d$-seperation of $A \perp_{\mathcal{G}} L^p \mid X^p$ and thus also the statistical conditional independence $A \perp\!\!\!\perp L^p \mid X^p$. This also gives us that $\mathbb{1}[A = \varnothing] \perp\!\!\!\perp L^p \mid X^p$, as the indicator is function of $A$. Proposition 5.1 then tells us immediately that the conditional distribution stays the same for $\widehat{L}(x) = 1$, i.e.

$$h^p(x) = P(L^p = 1 \mid X^p = x) = P(L = 1 \mid X = x) = h(x).$$

$\qquad \square$

### D.3 Invertible aggregated noise assumption and proofs

We restate the assumption and stability results again for completeness sake.

**Assumption 5.6** (**Marginalization with Invertible Aggregated Noise**). For an SCM with structural equations $f_E$ and $f_Y$, we assume that

    (i) there exists an aggregation function $b_{\mathrm{agg}}$ and a structural equation $f_{\mathrm{agg}}$ such that we can rewrite the composition $f_E(f_Y(x_C, u_Y), x_S, u_E)$ as $f_{\mathrm{agg}}(x_C, x_S, b_{\mathrm{agg}}(u_Y, u_E))$, and

    (ii) $f_{\mathrm{agg}}$ is invertible, that is, there exists $f_{\mathrm{agg}}^{-1}$ such that $f_{\mathrm{agg}}^{-1}(x_C, x_E, x_S) = b_{\mathrm{agg}}(u_E, u_Y)$.

**Linear Additive and Multiplicative SCMs satisfy Assumption 5.6** In the Section D we mention that Linear Additive models and Multiplicative SCMs satisfy Assumption 5.6. Here we will quickly show that this is the case. In general, the structural equations for the effect variables $X_E$ and $Y$ are given by the following structural equations:

$$Y = l_Y(X_C) + U_Y$$
$$X_E = l_E(Y, X_C, X_S) + U_E,.$$

where $l_Y$ and $l_E$ are linear functions (for example, $l_E(Y, X_C, X_S) = \beta_Y Y + \beta_C + X_C + \beta_S + X_S + \beta_0$). By substituting the expression of $Y$ into the equation for $X_E$ we get

$$X_E = l_E(l_Y(X_C) + U_Y, X_C, X_S) + U_E$$
$$= l_E(l_Y(X_C), X_C, X_S) + \beta U_Y + U_E.$$

We can rewrite the structural equations as

$$b_{\mathrm{agg}}(x) = \beta U_Y + U_E$$
$$f_{\mathrm{agg}}(x_C, x_S, u_{\mathrm{agg}}) = l_E(l_Y(x_C), x_C, x_S) + u_{\mathrm{agg}}$$
$$f_{\mathrm{agg}}^{-1}(x_C, x_S, x_E) = x_E - l_E(l_Y(x_C), x_C, x_S).$$

and thus Assumption 5.6 is satisfied.

In multiplicative settings with multilinear aggregation functions $m$ (linear in each argument, for example $m_E(X_C, Y, X_S) = X_C Y X_S \beta$), we get

$$Y = m_Y(X_C) U_Y$$
$$X_E = m_E(X_C, Y, X_S) U_E$$

Substituting the expression for $Y$ we get

$$X_E = m_E(X_C, Y, X_S) U_E$$
$$= m_E(X_C, m_Y(X_C) U_Y, X_S) U_E$$
$$= m_E(X_C, m_Y(X_C), X_S) U_Y U_E.$$

We can rewrite the structural equations as

$$u_{\mathrm{agg}} = b_{\mathrm{agg}}(u) := (U_Y U_E)$$
$$f_{\mathrm{agg}}(x_C, x_S, u_{\mathrm{agg}}) := m_E(x_C, m_Y(x_C), x_S) u_{\mathrm{agg}}$$
$$f_{\mathrm{agg}}^{-1}(x_C, x_S, x_E) = \frac{x_E}{m_E(x_C, m_Y(x_C), x_S)},$$

and thus Assumption 5.6 is satisfied.

**Theorem 5.7** (**Assumption 5.6 restores performative validity.**). *Let Assumption 5.6 be satisfied. Further assume that the unobserved causal influences stay the same post-recourse, that is $(U_Y, U_E) = (U_Y^p, U_E^p)$. Then, recourse methods that abstain from interventions on effects are performatively valid.*

*Proof.* We will again use the characterization given in Proposition 5.1. Using Assumption 5.6, we can show that the required conditional independence is present. Let $I_A$ be the index set on which an intervention $A$ is performed. We write,

$$P(L^p, I_A = \varnothing \mid X^p = x^p) = P(L^p \mid I_A = \varnothing, X^p = x^p) P(I = \varnothing \mid X^p = x^p).$$

We can rewrite the first probability as

$$P(L^p \mid I_A = \varnothing, X^p = x^p) = \int_{\mathcal{U}_Y} P(L^p = 1 \mid x^p, u_Y, I_A = \varnothing) p(u_Y \mid x^p, I_A = \varnothing) \, \mathrm{d}u_Y$$

$$= \int_{\mathcal{U}_Y} P(L^p = 1 \mid x^p_C, u_Y) p(u_Y \mid x^p, I_A = \varnothing) \, \mathrm{d}u_Y.$$

Where the second equality follows from the fact that $L^p$ is completely determined by $X^p_C$ and $U_Y$. For the conditional probability on the noise we have the following sequence of equalities, because we can condition on no intervention being performed

$$p(u_Y \mid X^p = x^p, I_A = \varnothing) = p(u_Y \mid X = x^p, I_A = \varnothing) = p(u_Y \mid X = x^p).$$

The first equality here follows from the fact that $X^p$ and $X$ have the same functional equations, when no intervention is performed. The last equality follows as $I_A$ is deterministic function of $X$.

What rests to show is that $p(u_Y \mid X^p = x^p) = p(u_Y \mid X = x^p)$. The basis for this proof is that $X_C, X_E$ and $X^p_C, X^p_E$ constrain $U = (U_Y, U_E)$ in the same way, whenever no interventions on the effects are performed. That is, for $p(x, x^p) > 0$ it holds that either the conditionals $p(x_E \mid x_C, x_S, u_Y, u_E)$ and $p(x^p_E \mid x^p_C, x^p_S, U_Y, U_E)$ are exactly zero or exactly one. The reason is Assumption 5.6 on the structural equations. We know that

$$p(x_E \mid x_C, x_S, u_Y, u_E) = \mathbb{1}[x_E = f_s(x_C, x_S, b_s(u_Y, u_E))]$$
$$p(x^p_E \mid x^p_C, x^p_S, u_Y, u_E) = \mathbb{1}[x^p_E = f_s(x^p_C, x^p_S, b_s(u_Y, u_E))].$$

We can reformulate the constraints as

$$b_s(u_Y, u_E) = f_s^{-1}(x_C, x_E, x_S)$$
$$b_s(u_Y, u_E) = f_s^{-1}(x^p_C, x^p_E, x^p_S).$$

We recall that

$$p(u_Y, u_E \mid X = x) = \frac{p(u_Y, u_E, x)}{p(x)}$$

and that

$$p(u_Y, u_E \mid X^p = x^p) = \frac{p(u_Y, u_E, x^p)}{p(x^p)}.$$

The joint distribution $p(u_Y, u_E, x)$, can be written out as

$$p(u_Y, u_E, x) = p(u_Y)p(u_E)p(x_C)p(x_S)p(x_E \mid x_C, x_S, u_Y, u_E),$$

giving a conditional distribution of

$$p(u_Y, u_E \mid X = x) = \frac{p(u_Y)p(u_E)p(x_C)p(x_S)p(x_E \mid x_C, x_S, u_Y, u_E)}{\int_{\mathcal{U}} p(u'_Y)p(u'_E)p(x_C)p(x_S)p(x_E \mid x_C, x_S, u'_Y, u'_E) \, \mathrm{d}u'}$$

$$= \frac{p(u_Y)p(u_E)p(x_E \mid x_C, x_S, u_Y, u_E)}{\int_{\mathcal{U}} p(u'_Y)p(u'_E)p(x_E \mid x_C, x_S, u'_Y, u'_E) \, \mathrm{d}u'}. \tag{3}$$

Now, for the second conditional we can write it as

$$p(u_Y, u_E, x^p) = \int_{\mathcal{X}_C} p(u_Y, u_E, x^p, x_C) \, \mathrm{d}x_C$$

$$= \int_{\mathcal{X}_C} p(u_Y, u_E, x_C) p(x^p_C \mid x_C, u_Y, u_E) p(x^p_E, x^p_S \mid x^p_C, u_Y, u_E) \, \mathrm{d}x_C$$

$$= p(u_Y)p(u_E)p(x^p_E, x^p_S \mid x^p_C, u_Y, u_E) \int_{\mathcal{X}_C} p(x_C) p(x^p_C \mid x_C, u_Y, u_E) \, \mathrm{d}x_C$$

$$= p(u_Y)p(u_E)p(x^p_E, x^p_S \mid x^p_C, u_Y, u_E) \int_{\mathcal{X}_C} p(x_C) p(x^p_C \mid x_C, u_s = b_s(u_Y, u_E)) \, \mathrm{d}x_C$$

Because of the assumption, we know that $p(x_E^p, x_S^p \mid x_C^p, u_Y, u_E)$ is zero unless $b_s(u_Y, u_E) = f_s^{-1}(x_C^p, x_E^p, x_S^p)$. Thus we can replace $p(x_C^p \mid x_C, u_s = b_s(u_Y, u_E))$ with $p(x_C^p \mid x_C, u_s = f_s^{-1}(x_C^p, x_E^p, x_S^p))$ and get

$$p(u_Y, u_E, x^p) = p(u_Y)p(u_E)p(x_E^p, x_S^p \mid x_C^p, u_Y, u_E) \int_{\mathcal{X}_C} p(x_C)p(x_C^p \mid x_C, u_s = f_s^{-1}(x^p))\, \mathrm{d}x_C.$$

Remark that the final integral only depends on $x^p$, we will shorten this integral by setting

$$Z(x^p) = \int_{\mathcal{X}_C} p(x_C)p(x_C^p \mid x_C, u_s = f_s^{-1}(x^p))\, \mathrm{d}x_C.$$

Plugging everything together we get

$$
\begin{aligned}
p(u_Y, u_E \mid X^p = x^p) &= \frac{p(u_Y)p(u_E)p(x_E^p, x_S^p \mid x_C^p, u_Y, u_E)Z(x^p)}{\int_{\mathcal{U}} p(u_Y')P(u_E')p(x_E^p, x_S^p \mid x_C^p, u_Y', u_E')Z(x^p)\, \mathrm{d}u'} \\
&= \frac{p(u_Y)p(u_E)p(x_S^p)p(x_E^p \mid x_C^p, x_S^p, u_Y, u_E)Z(x^p)}{\int_{\mathcal{U}} p(u_Y')P(u_E')p(x_S^p)p(x_E^p, \mid x_C^p, x_S^p, u_Y', u_E')Z(x^p)\, \mathrm{d}u'} \\
&= \frac{p(u_Y)p(u_E)p(x_E^p \mid x_C^p, x_S^p, u_Y, u_E)}{\int_{\mathcal{U}} p(u_Y')p(u_E')p(x_E^p \mid x_C^p, x_S^p, u_Y', u_E')\, \mathrm{d}u'}.
\end{aligned}
\tag{4}
$$

If we now substitute $x = x^p$ into Equation (3), we observe that the expressions in Equations (3) and (4) are equal. $\qquad\square$

**Theorem D.1.** *Consider the causal graph in Figure 2(a). Assume that $U = U^p$ and that Assumption 5.6 is satisfied. If the recourse method suggests interventions on direct effects with non-zero probability, then there exist $x$ with $\widehat{L}(x) = 1$ for which $h^p(x) \neq h(x)$ and the post-recourse conditional probability is described by*

$$h^p(x) = \alpha(x)h(x) + \beta(x)g(x) + \gamma(x)r(x),$$

*where $g(x) \leq t_c$, $r(x) \in [0, 1]$ and $\alpha, \beta, \gamma$ are all functions bounded in $[0, 1]$.*

Compared to Lemma E.1, one of the components will always be smaller than the decision threshold, but there will also be a component, for which it is impossible to tell if it is higher or lower than the decision threshold in general.

*Proof.* We will prove this result by a direct calculation. First, we split the post-recourse conditional probability by conditioning on where the intervention are performed. Let $\mathcal{I}$ be all possible combinations of the sets $C, E, S$ and the empty set. We can rewrite the probability as,

$$P(L^p = 1 \mid X^p = x^p) = \sum_{i \in \mathcal{I}} P(I_A = i \mid X^p = x)P(L^p = 1 \mid X^p = x, I_A = i).$$

We immediately have that $P(L^p = 1 \mid X^p = x, I_A = \varnothing) = P(L = 1 \mid X = x) = h(x)$ and for any $i$ without $E$ we have $P(L^p = 1 \mid X^p = x, I_A = i) = P(L = 1 \mid X = x) = h(x)$ by the proof of Theorem 5.7. This gives us that $\alpha = \sum_{i \in \mathcal{I}:\ E \notin i} P(I_A = i \mid X^p = x)$

As the next step, we will focus on the terms with $E \in i$, but $C \notin i$. This will be the $g$ function and we set

$$g(x) := \sum_{i \in \mathcal{I}:\ E \in i, C \notin i} P(Y^p = 1 \mid X^p = x, I_A = i)$$

$$\beta(x) := \sum_{i \in \mathcal{I}:\ E \in i, C \notin i} P(I_A = i \mid X^p = x).$$

We show by a direct calculation that each term can be bounded by the pre-recourse probability. We make use of several conditional independencies in the graph depicted in Figure 2(b) and the fact that

the structural equation of $Y$ is the same as the structural equation of $Y^p$.

$$P(L^p = 1 \mid x^p, i) = \int_{\mathcal{X}} P(L^p = 1 \mid x^p, x, i) p(x \mid x^p, i) \, \mathrm{d}x$$

$$= \int_{\mathcal{X}} \int_{\mathcal{U}_Y} P(L^p = 1 \mid x^p, x, u_Y, i) p(u_Y \mid x^p, x, i) p(x \mid x^p, i) \, \mathrm{d}u_Y \, \mathrm{d}x$$

$$= \int_{\mathcal{X}} \int_{\mathcal{U}_Y} P(L = 1 \mid X = x_C^p, u_Y) p(u_Y \mid x^p, x, i) p(x \mid x^p, i) \, \mathrm{d}u_Y \, \mathrm{d}x$$

$$= \int_{\mathcal{X}} \int_{\{u_Y \,:\, f_Y(x_C^p, u_Y) \geq 0\}} p(u_Y \mid x^p, x, i) p(x \mid x^p, i) \, \mathrm{d}u_Y \, \mathrm{d}x.$$

Now, we use that $U_Y \perp_{\mathcal{G}} X^p, A \mid X \implies U_Y \perp\!\!\!\perp X^p, A \mid X$ and that $x_C^p = x_C$, as only the effects are intervened upon, to write

$$P(L^p = 1 \mid X^p = x^p, i) = \int_{\mathcal{X}} p(x \mid x^p, i) \left[ \int_{\{u_Y \,:\, f_Y(x_C, u_Y) \geq 0\}} p(u_Y \mid x) \, \mathrm{d}u_Y \right] \mathrm{d}x$$

$$= \int_{\mathcal{X}} p(x \mid x^p, i) P(L = 1 \mid X = x) \, \mathrm{d}x$$

$$\leq t_c \cdot \int_{\mathcal{X}} p(x \mid x^p, i) \, \mathrm{d}x$$

$$= t_c.$$

Where the last inequality is a consequence of only suggesting an intervention whenever $h(x) < t_c$ and thus $p(x \mid x^p, i)$ will only be non-zero for points $x$ with $h(x) < t_c$. The inequality will be strict if $P(I_A \cap i \neq \varnothing \mid X^p = x^p) > 0$.

The final term will be the $r$-function, which is defined as

$$r(x) := \sum_{i \in \mathcal{I} \,:\, E \in i, C \in i} P(L^p = 1 \mid X^p = x^p, I_A = i).$$

We can repeat the previous derivation, but we do not use the step $x_C^p = x_C$. This gives

$$r(x^p) = \int_{\mathcal{X}} p(x \mid x^p, (C, E)) \left[ \int_{\{u_Y \,:\, f_Y(x_C^p, u_Y) \geq 0\}} p(u_Y \mid x) \, \mathrm{d}u_Y \right] \mathrm{d}x.$$

Now that we cannot change $x_C^p = x_C$, this expression cannot be simplified further. The relation between $u_Y$ and $x$ could have a positive influence or a negative influence on the overall probability.

Putting everything together gives the required expression. $\square$

## E  Proofs of Section 5.2

Now, we will perform the calculation to get the exact form of the conditional distribution when the noise is resampled.

**Proposition E.1.** *Consider the causal graph in Figure 2(a). Let Assumption 5.4 be satisfied. Further assume that the probability of interventions on direct effects is non-zero. Then, there exist $x$ with $\widehat{L}(x) = 1$ for which $h^p(x) \neq h(x)$ and the post-recourse distribution takes the following form,*

$$h^p(x) = (1 - \beta(x))h(x) + \beta(x)P(L = 1 \mid X_C = x_C) \quad \beta(x) \in (0, 1]$$

*Proof.* To calculate the exact form of the conditional, we write $I_A$ for the index set of the recourse recommendation and we condition on which interventions are performed and splitting up the probabilities. As before, $\mathcal{I}$ will denote the set of all combinations of $C, E, S$ and the empty set. We write the conditional probability as

$$P(L^p = 1 \mid X^p = x) = \sum_{i \in \mathcal{I}} P(I_A = i \mid X^p = x)P(L^p = 1 \mid X^p = x, I_A = i) \qquad (5)$$

Whenever there was no intervention was performed, the conditional distribution does not change as all the noise distributions are the same and the structural equations remain unchanged. This means that

$$P(L^p = 1 \mid X^p = x, I_A = \varnothing) = P(L = 1 \mid X = x, I_A = \varnothing) = P(L = 1 \mid X = x).$$

Whenever an intervention on the cause or one of the spouses is performed is performed, no change will be observed. So without loss of generality we can focus only on the case where $I_A = E$.

Now that we assume that an intervention is performed on the effects of $Y^p$ only, we see that the arrow from $Y^p$ to $X_E^p$ and the arrow from $X_S^p$ to $X_E^p$ area broken. Coincidentally, the variables $Y^p$ and $L^p$ only depend on $X_C^p$.

$$\begin{aligned} P(L^p = 1 \mid I_A = E, X^p = x) &= P(L^p = 1 \mid I_A = E, X_C^p = x_C) \\ &= P(L^p = 1 \mid X_C^p = x_C) \\ &= P(L = 1 \mid X_C = x_C). \end{aligned}$$

Combining everything together we find that Equation (5) reduces to

$$\begin{aligned} P(L^p = 1 \mid X^p = x) &= \sum_{i \in \mathcal{I}:\, E \notin i} P(I_A = i \mid X^p = x)P(L = 1 \mid X = x) \\ &\quad + \sum_{i \in \mathcal{I}:\, E \in i} P(I_A = i \mid X^p = x)P(L = 1 \mid X_C = x_C) \\ &= (1 - \beta(x))P(L = 1 \mid X = x) + \beta(x)P(L = 1 \mid X_C = x_C). \end{aligned}$$

Where we set

$$\beta(x) = \sum_{i \in \mathcal{I}:\, E \in i} P(I_A = i \mid X^p = x)$$

and hence $\beta \in (0, 1]$. $\qquad \square$

All the results in this section and the previous section now give Corollary 5.9.

**Corollary 5.9.** *In a setting where the first source of invalidity (influence on $A$) can be excluded, that is if either Assumption 5.4 or 5.6 holds, recourse algorithms that abstain from interventions on direct effects are performatively valid but other recourse algorithms may not. Without additional assumptions, **performative validity can only be guaranteed for ICR, but not for CR and CE**.*

*Proof.* Theorem 5.7 gives us that ICR is performatively stable when the noise stays the same pre- and post-recourse and Proposition 5.5 tells us the same when the noise is independently resampled. Propositions D.1 and E.1 tell us that CR and CE may not be performatively valid. $\qquad \square$

### E.1  Details of Example 5.8

**Interventions on effects cause performative invalidity**    First, we will calculate the explicit form of the conditional distribution. That is, $h(x) = P(L = 1 \mid X = x) = \Phi\left((x_C + x_E)/\sqrt{2}\right)$, where $\Phi$ is the cumulative distribution function of the standard normal distribution. We start by rewriting the conditional distribution as

$$\begin{aligned} P(L = 1 \mid X = x) &= P(Y \geq 0 \mid X = x) \\ &= \int_0^\infty p(y \mid x)\, \mathrm{d}y. \end{aligned}$$

An explicit expression for the conditional density $p(y \mid x)$ can be obtained through a Markov decomposition using the structure of the graph and an application of the Bayes' rule

$$p(y \mid x) = \frac{p(x_C, x_E, y)}{p(x_C, x_E)}$$

$$\text{(Markov decomposition)} = \frac{p(x_C)p(y \mid x_C)p(x_E \mid y)}{p(x_C)p(x_E \mid x_C)}$$

$$\text{(Bayes' rule)} = \frac{p(y \mid x_C)p(x_E \mid y)}{p(x_E \mid x_C)}. \tag{6}$$

Because we know the exact relations between the variables through the structural equations, we can infer the conditional densities needed in Equation (6). Remark that the noise variables $U_Y$ and $U_Y$ are distributed according to independent standard normals. The structural equation for $Y$ is defined as $f_Y(X_C, U_Y) = X_C + U_Y$, und since $U_Y$ and $X_C$ are independent we get that $Y = x_c + U_Y$ when we condition on $X_C = x_c$, meaning that $P(Y|X_C) \sim N(x_c, 1)$ which is the distribution of $U_Y$ shifted by $x_C$. We can repeat this derivation for the other variables to conclude that

$$Y \mid X_C = x_C \sim \mathcal{N}(x_C, 1),$$
$$X_E \mid Y = y \sim \mathcal{N}(y, 1)$$
$$X_E \mid X_C = x_C \sim \mathcal{N}(x_C, 2).$$

Substituting the densities of these random variables into Equation (6) gives

$$p(y \mid x) = \frac{\sqrt{\frac{1}{2\pi}}e^{-\frac{1}{2}(y - x_c)^2}\sqrt{\frac{1}{2\pi}}e^{-\frac{1}{2}(x_E - y)^2}}{\sqrt{\frac{1}{4\pi}}e^{-\frac{1}{4}(x_E - x_C)^2}}$$

$$= \sqrt{\frac{1}{\pi}}\frac{e^{-\frac{1}{2}y^2 + x_c y - \frac{1}{2}x_c^2 - \frac{1}{2}x_E^2 + x_E y - \frac{1}{2}y^2}}{e^{-\frac{1}{4}x_E^2 + \frac{1}{2}x_E x_C - \frac{1}{4}x_C^2}}$$

$$= \sqrt{\frac{1}{\pi}}e^{-y^2 + (x_C + x_E)y - \frac{1}{4}x_c^2 - \frac{1}{4}x_E^2 - \frac{1}{2}x_E x_C}$$

$$= \sqrt{\frac{1}{\pi}}e^{-\left(y - \frac{x_C + x_E}{2}\right)^2}.$$

The last expression we recognize as the density of a $\mathcal{N}\left(\frac{x_C + x_E}{2}, \frac{1}{2}\right)$ distribution. Finally, we use that we can translate and rescale the cdf of a normal distribution to the cdf of the standard normal. Let $\Phi_{\mu, \sigma^2}$ be the cdf of a normal distribution, then it holds that

$$\Phi_{\mu, \sigma^2}(x) = \Phi\left(\frac{x - \mu}{\sigma}\right).$$

Using $\mu = \frac{x_C + x_E}{2}$ and $\sigma^2 = \frac{1}{2}$, we get

$$P(L = 1 \mid X = x) = P(Y \geq 0 \mid X = x)$$
$$= 1 - P(Y \leq 0 \mid X = x)$$
$$= 1 - \Phi\left(\frac{0 - \frac{x_C + x_E}{2}}{\sqrt{1/2}}\right)$$
$$= \Phi\left((x_C + x_E)/\sqrt{2}\right).$$

The final equality follows from the symmetry of the standard normal distribution around 0.

The Bayes optimal classifier is given by $\widehat{L}(x) = \mathbb{1}\left[P(L = 1 \mid X = x) \geq \frac{1}{2}\right]$. In this case, we have the threshold $t_c = \frac{1}{2}$ and the decision boundary is at those points where

$$\Phi\left((x_C + x_E)/\sqrt{2}\right) = \frac{1}{2} \iff (x_C + x_E)/\sqrt{2} = 0 \iff x_C = -x_E.$$

So, in the pre-recourse setting, the decision boundary is at $x_C = -x_E$.

Now we apply Proposition E.1, to get the expression for the post-recourse conditional distribution, which is

$$h^p(x) = (1 - \beta(x))(x)h(x) + \beta(x)P(L = 1 \mid X_C = x_C).$$

Similarly as before, we can calculate that $P(L = 1 \mid X_C = x_C) = \Phi(x_C)$. The property to note is that $\Phi(x_C) < \frac{1}{2}$ whenever $x_C < 0$. Now, for any point with $x_C < 0$, but $x_C = -x_E$ we observe

$$h^p(x) = (1 - \beta(x))h(x) + \beta(x)P(L = 1 \mid X_C = x_c) < \frac{\alpha}{2} + \frac{\beta}{2} = \frac{1}{2}.$$

Here, the $\beta(x)$ indicates the proportion of the post-recourse individuals that intervened on the effect variables, which will be a non-zero amount in this example. The fact that $h^p(x) < \frac{1}{2}$, but $h(x) = \frac{1}{2}$ shows that all points on the decision boundary with $x_C < 0$, are invalidated post-recourse. Indeed, the final mixed distribution will be given by

$$h^m(x) = \alpha h(x) + (1 - \alpha)h^p(x) < \frac{\alpha}{2} + \frac{1 - \alpha}{2} = \frac{1}{2},$$

where $\alpha$ indicates the mixing parameter.

Finally, the probability mass of the people that get rejected that were originally accepted is non-zero, because the recourse recommendation moves a non-zero mass towards the decision boundary.

## F   Experiments

All code is publicly available via GitHub[1].

### F.1   Settings

In our experiments, we report the results for five synthetic and one real-world dataset. In the synthetic settings we focus on varying the type of functional relationship while ensuring that the data generating process has finite support. Furthermore, we chose the parameters such that the prediction of the corresponding model can always be changed by modifying only one of the features.

To obtain data with finite support, we rely on the discrete binomial distribution. The binomial distribution normally takes two parameters, $n$ and $p$, where $n$ is the number of trials and $p$ the probability of a positive outcome in each trial. The support of the binomial are integers in $\{0, \ldots, n\}$ and the mean is $np$.

For our settings, we sometimes shift the distribution by an offset, e.g., to ensure positive values. We refer to this shifted binomial as $\mathrm{ShBin}(n, p, \mu)$ where $\mu$ is the new mean. To sample from $\mathrm{ShBin}(n, p, \mu)$, we sample values from $\mathrm{Bin}(n, p)$ and shift them by the offset $\mu - np$.

Now we are ready to introduce all synthetic DGPs in detail.

**Setting F.1 (Additive Noise, LinAdd).**

$$
\begin{aligned}
X_C &:= U_C & U_C &\sim \mathrm{ShBin}(8, 0.5, 0) \\
Y &:= X_C + U_Y & U_Y &\sim \mathrm{ShBin}(2, 0.5, 0) \\
X_E &:= Y + X_C + U_E & U_E &\sim \mathrm{ShBin}(2, 0.5, 0)
\end{aligned}
$$

**Setting F.2 (Multiplicative Noise, LinMult).**

$$
\begin{aligned}
X_C &:= U_C & U_C &\sim \mathrm{ShBin}(5, 0.5, 5/2 + 1) \\
Y &:= X_C U_Y & U_Y &\sim \mathrm{ShBin}(1, 0.5, 0.5 + 1) \\
X_E &:= Y X_C U_E & U_E &\sim \mathrm{ShBin}(1, 0.5, 0.5 + 1)
\end{aligned}
$$

---

[1]https://github.com/gcskoenig/performative-recourse-experiments

**Setting F.3 (Nonlinear Additive, NlinAdd).**

$$X_C := U_C \qquad\qquad\qquad U_C \sim \text{ShBin}(8, 0.5, 0)$$
$$Y := (X_C)^2 + U_Y \qquad\qquad U_Y \sim \text{ShBin}(2, 0.5, 0)$$
$$X_E := (Y + X_C)^2 + U_E \qquad U_E \sim \text{ShBin}(2, 0.5, 0)$$

**Setting F.4 (Nonlinear Multiplicative, NlinMult).**

$$X_C := U_C \qquad\qquad\qquad U_C \sim 0.5\text{ShBin}(2, 0.5, 2) + 0.5\text{ShBin}(4, 0.5, 4)$$
$$Y := X_C^2 U_Y \qquad\qquad\qquad U_Y \sim \text{ShBin}(2, 0.5, 2)$$
$$X_E := (Y + X_C)^2 U_E \qquad\quad U_E \sim \text{ShBin}(2, 0.5, 2)$$

**Setting F.5 (Polynomial Noise, LinCubic).**

$$X_C := U_C \qquad\qquad\qquad U_C \sim \text{ShBin}(4, 0.5, -1)$$
$$Y := (X_C + U_Y)^3 \qquad\qquad U_Y \sim \text{ShBin}(2, 0.5, 1)$$
$$X_E := (X_C + U_E - Y)^3 \qquad U_E \sim \text{ShBin}(1, 0.5, 0.5)$$

For the *GPA* example we obtain data from [OpenIntro, 2020] and assume the causal graph suggested in [Harris et al., 2022]. Then, we fit a linear Gaussian SCM, that is, for each node we fit a linear model to predict the node from its parents, and fit a normal distribution to the residual to obtain the corresponding noise distribution. For the *Credit* example we obtain data from Chen et al. [2023], and assume the causal graph suggested by [Yeh, 2009]. Then, we fit a random forest based additive Gaussian SCM, meaning that we fit a random forest to predict each node from its parents, and fit a normal distribution on the residuals to obtain the noise distributions.

**Setting F.6 (College Admission, SAT).**

$$X_C := U_C \qquad\qquad\qquad U_C \sim \mathcal{N}(0, 1)$$
$$Y := \alpha_Y X_C + U_Y \qquad\qquad U_Y \sim \mathcal{N}(\mu_Y, \sigma_Y)$$
$$X_E := \alpha_E Y + U_E \qquad\qquad U_E \sim \mathcal{N}(\mu_Y, \sigma_Y)$$

In all settings, the target variable $Y$ is binarized using the median as treshold. That is,

$$L := \mathbb{1}[Y \geq \text{median}(Y)].$$

**Costs** In our experiments we want to compare the different recourse methods, and to reveal their differences we chose the cost such that interventions on effects are more lucrative. Then CE and CR intervene on the effects, and ICR intervenes on the causes. To do so, we define the cost functions for CR and ICR as

$$cost(x, \text{do}(a)) = \sum_{j \in I_a} (x_j - \theta_j)^2 \gamma_j$$

and for CE as

$$cost(x, x') = \sum_j (x_j - x'_j)^2 \gamma_j.$$

The weight $\gamma_j$ ensures that interventions on effects are more lucrative. It is defined as

$$\gamma_j := \frac{1}{\pi_j \sigma_j^2},$$

where $\pi_j$ is the index of the node when they are (partially) ordered according to the causal graph, that is, causes recieve lower indices than their descendants and therfore more weight in the cost. Further, we normalize the cost using each feature's variance $\sigma_j^2$.

### F.2 Experiment setup

**Random seeds** We conducted each experiment 10 times with seeds $40, \ldots, 49$. In Figure 3 (left), we report averages over all runs, that is, the average minimum difference in conditional distribution, the average maximum difference in conditional distribution, and the average expected difference in conditional distribution. In Figure 3 (right) the dots represent averages over 10 runs and the errorbars the standard deviation $\sigma$ (the expected value $\pm\sigma$). In Table 1 we report the mean and standard deviation for each metric.

**Decision model** We use a decision tree with default hyperparamters for the discrete settings and a logistic regression with default hyperparamters in the real-world setting. We use `sklearn` to fit the model. To ensure that the original model is as accurate as possible we sample $10^5$ fresh datapoints as an independent training set.

**Sampling from the post-recourse distribution** Given the SCM and the decision model, we are ready to compute recourse recommendations. Therefore we sample fresh data and randomly pick 5000 rejected observations in the simulation settings, and 1000 rejected observations for the real-world setting, for which we then generate recourse recommendations. The detailed procedures for generating recourse are explained below. Given the recourse recommendation, we compute the true post-recourse outcomes for the individuals. Therefore, we exploit that we sampled the data ourselves, meaning that we have access to the unobserved causal influences that determine the observations. In other words, we are able to compute the ground-truth outcomes for the respective recommendation. Specifically, for each individual, we fix the unobserved causal influences to the respective values, and then replace the structural equations according to the interventions. Then, we determine the new characteristics of the applicant based on the unobserved causal influences and the structural equations. Based on these ground-truth samples from the post-recourse distribution of rejected applicants we assess Q1 and Q2. We note that sampling from the post-recourse distribution is expensive, since each post-recourse sample requires solving the recourse optimization problem.

**Q1: How did the conditional distribution change?** To quantify the change in distribution, we compute the conditional probability of a favorable label $L = 1$ both in the original distribution and in the part of the distribution that may have changed as a result of recourse. More formally, we compute $P(L^p = 1 \mid X^p = x, \widehat{L} = 1)$ and $P(L = 1 \mid X = x)$. Notably, we do not take $P(L^p = 1 \mid X^p = x, \widehat{L} = 0)$ into account, since for the subpopulation the conditional distribution must be the same pre and post-recourse, that is $P(L^p = 1 \mid X^p = x, \widehat{L} = 0) = P(L = 1 \mid X = x)$. To make the estimates as accurate as possible, we sample many observations (5000) and designed the DGPs such that the number of possible values is small, and in each bucket enough samples can be found. We aggregate the point-wise differences using the min, max, and expected value. To ensure that buckets with larger sample size get more weight in the expected value, and to reflect the perspective of recourse implementing individuals, we weigh the points according to $P(X^p = x \mid \widehat{L} = 0)$.

**Q2: Does the shift impact acceptance rates?** To obtain the updated model and evaluate the impact of the update on acceptance rates, we split the sample of recourse implementing individuals in half. The first half is used to fit the updated model, the second half to evaluate the respective acceptance rate. To fit the updated model, the sample of recourse implementing individuals is matched with their respective pre-recourse observations as well as a sample of accepted applicants of the same size. As a result, the updated model is fitted on one third post-recourse samples and two thirds pre-recourse samples. To compute the new acceptance rate, we take the other half of the post-recourse samples and compute the respective decisions with respect to the original and the updated model.

### F.3 Detailed results

The detailed results are reported in Table 1.

Furthermore, we report model accuracies in Table 2. We note that the model accuracy may drop even if the model remains optimal post-recourse. The reason is that recourse commonly moves data closer to the decision boundary, where model uncertainty is higher, as pointed out in [Fokkema et al., 2024].

Table 1: **Detailed Experiment Results.** We report all detailed results. For each outcome, we report the mean and standard deviation over the 10 runs. The outcomes are the expected, maximum, and minimum difference in conditional distribution, as well as the difference in acceptance rate.

| Method | Setting | Exp. Dist. Diff. mean | std | Max. Dist. Diff. mean | std | Min. Dist. Diff. mean | std | Acc. Rate Diff. mean | std |
|---|---|---|---|---|---|---|---|---|---|
| ind. ICR | LAdd | 0.00 | ±0.01 | 0.02 | ±0.05 | 0.00 | ±0.00 | 0.00 | ±0.00 |
| | LMult | 0.00 | ±0.00 | 0.00 | ±0.00 | 0.00 | ±0.00 | 0.00 | ±0.00 |
| | NLAdd | 0.00 | ±0.00 | 0.00 | ±0.00 | 0.00 | ±0.00 | 0.00 | ±0.00 |
| | NLMult | 0.00 | ±0.00 | 0.00 | ±0.00 | 0.00 | ±0.00 | 0.00 | ±0.00 |
| | LCubic | 0.21 | ±0.00 | 0.67 | ±0.00 | 0.00 | ±0.00 | 0.00 | ±0.00 |
| | GPA | − | − | − | − | − | − | 0.00 | ±0.00 |
| | Credit | − | − | − | − | − | − | 0.01 | ±0.01 |
| sub. ICR | LAdd | 0.00 | ±0.00 | 0.00 | ±0.00 | 0.00 | ±0.00 | 0.00 | ±0.00 |
| | LMult | 0.00 | ±0.00 | 0.00 | ±0.00 | 0.00 | ±0.00 | 0.00 | ±0.00 |
| | NLAdd | 0.00 | ±0.00 | 0.00 | ±0.00 | 0.00 | ±0.00 | 0.00 | ±0.00 |
| | NLMult | 0.00 | ±0.00 | 0.00 | ±0.00 | 0.00 | ±0.00 | 0.00 | ±0.00 |
| | LCubic | 0.00 | ±0.00 | 0.00 | ±0.00 | 0.00 | ±0.00 | 0.00 | ±0.00 |
| | GPA | − | − | − | − | − | − | 0.00 | ±0.00 |
| | Credit | − | − | − | − | − | − | 0.01 | ±0.01 |
| ind. CR | LAdd | −0.68 | ±0.13 | −0.46 | ±0.04 | −0.95 | ±0.08 | −0.93 | ±0.10 |
| | LMult | −0.65 | ±0.12 | −0.55 | ±0.08 | −0.85 | ±0.18 | −0.85 | ±0.18 |
| | NLAdd | −0.97 | ±0.03 | −0.72 | ±0.36 | −1.00 | ±0.00 | −0.87 | ±0.02 |
| | NLMult | −0.12 | ±0.00 | 0.00 | ±0.00 | −0.49 | ±0.02 | −0.16 | ±0.01 |
| | LCubic | −0.57 | ±0.01 | 0.00 | ±0.00 | −1.00 | ±0.00 | −0.98 | ±0.00 |
| | GPA | − | − | − | − | − | − | −0.81 | ±0.05 |
| | Credit | − | − | − | − | − | − | −0.49 | ±0.03 |
| sub. CR | LAdd | −0.47 | ±0.06 | −0.13 | ±0.20 | −0.84 | ±0.06 | −0.90 | ±0.08 |
| | LMult | −0.61 | ±0.08 | −0.54 | ±0.07 | −0.76 | ±0.16 | −0.85 | ±0.18 |
| | NLAdd | −0.97 | ±0.04 | −0.72 | ±0.36 | −1.00 | ±0.00 | −0.87 | ±0.02 |
| | NLMult | −0.16 | ±0.01 | 0.00 | ±0.00 | −0.47 | ±0.01 | −0.01 | ±0.01 |
| | LCubic | −0.38 | ±0.03 | 0.00 | ±0.00 | −0.84 | ±0.01 | −0.98 | ±0.00 |
| | GPA | − | − | − | − | − | − | −0.79 | ±0.03 |
| | Credit | − | − | − | − | − | − | −0.39 | ±0.04 |
| CE | LAdd | −0.71 | ±0.13 | −0.46 | ±0.04 | −0.97 | ±0.07 | −0.95 | ±0.09 |
| | LMult | −0.65 | ±0.13 | −0.55 | ±0.08 | −0.82 | ±0.21 | −0.85 | ±0.18 |
| | NLAdd | −0.97 | ±0.04 | −0.71 | ±0.38 | −1.00 | ±0.00 | −0.88 | ±0.01 |
| | NLMult | −0.16 | ±0.01 | 0.00 | ±0.00 | −0.49 | ±0.02 | −0.01 | ±0.01 |
| | LCubic | −0.41 | ±0.01 | 0.00 | ±0.00 | −0.83 | ±0.01 | −0.98 | ±0.00 |
| | GPA | − | − | − | − | − | − | −0.81 | ±0.04 |
| | Credit | − | − | − | − | − | − | −0.47 | ±0.05 |

Table 2: **Model Accuracies.** We report the absolute model accuracies of the **o**riginal model and the **r**efit on both the original data (O and R) and post-recourse data (OP and RP). All accuracies are computed on test data and averaged over ten runs.

| SCM | Type | O | R | OP | RP |
|---|---|---|---|---|---|
| LAdd | CE | 0.91 | 0.90 | 0.51 | 0.89 |
| LAdd | sub. ICR | 0.91 | 0.91 | 0.96 | 0.96 |
| LAdd | ind. CR | 0.91 | 0.90 | 0.51 | 0.89 |
| LAdd | ind. ICR | 0.91 | 0.91 | 0.96 | 0.96 |
| LAdd | sub. CR | 0.91 | 0.90 | 0.53 | 0.89 |
| LMult | CE | 0.87 | 0.87 | 0.50 | 0.85 |
| LMult | sub. ICR | 0.87 | 0.87 | 0.94 | 0.94 |
| LMult | ind. ICR | 0.87 | 0.87 | 0.94 | 0.94 |
| LMult | ind. CR | 0.87 | 0.87 | 0.50 | 0.85 |
| LMult | sub. CR | 0.87 | 0.87 | 0.50 | 0.85 |
| NLAdd | sub. ICR | 0.91 | 0.91 | 1.00 | 0.99 |
| NLAdd | ind. ICR | 0.91 | 0.91 | 0.99 | 0.99 |
| NLAdd | sub. CR | 0.91 | 0.90 | 0.58 | 0.83 |
| NLAdd | ind. CR | 0.91 | 0.90 | 0.58 | 0.83 |
| NLAdd | CE | 0.91 | 0.89 | 0.58 | 0.83 |
| NLMult | ind. CR | 1.00 | 1.00 | 0.84 | 0.92 |
| NLMult | sub. ICR | 1.00 | 1.00 | 1.00 | 1.00 |
| NLMult | sub. CR | 1.00 | 1.00 | 0.84 | 0.83 |
| NLMult | CE | 1.00 | 1.00 | 0.84 | 0.84 |
| NLMult | ind. ICR | 1.00 | 1.00 | 1.00 | 1.00 |
| LCubic | sub. CR | 0.92 | 0.86 | 0.54 | 0.83 |
| LCubic | CE | 0.92 | 0.86 | 0.54 | 0.83 |
| LCubic | ind. CR | 0.92 | 0.86 | 0.55 | 0.83 |
| LCubic | ind. ICR | 0.92 | 0.92 | 0.95 | 0.95 |
| LCubic | sub. ICR | 0.92 | 0.92 | 0.96 | 0.96 |
| GPA | ind. CR | 0.73 | 0.70 | 0.51 | 0.68 |
| GPA | ind. ICR | 0.73 | 0.73 | 0.81 | 0.81 |
| GPA | CE | 0.73 | 0.72 | 0.51 | 0.70 |
| GPA | sub. CR | 0.73 | 0.72 | 0.51 | 0.69 |
| GPA | sub. ICR | 0.73 | 0.73 | 0.80 | 0.80 |
| Credit | CE | 0.60 | 0.58 | 0.57 | 0.60 |
| Credit | ind. CR | 0.61 | 0.59 | 0.57 | 0.59 |
| Credit | sub. CR | 0.62 | 0.59 | 0.61 | 0.62 |
| Credit | sub. ICR | 0.61 | 0.60 | 0.76 | 0.76 |
| Credit | ind. ICR | 0.61 | 0.59 | 0.76 | 0.76 |

## F.4 Estimation of improvement and acceptance rates

When searching for the optimal recourse recommendation, we have to evaluate the probability of a positive outcome for an action given the observation of the individual and given the available causal knowledge. As follows, we explain how this estimation is implemented in our code. The probability estimates are based on samples of size 1000.

**Estimating individualized outcomes** The individualized versions of CR and ICR are based on individualized effect estimation and assume knowledge of the SCM. To estimate individualized outcomes using a SCM, be it the acceptance or improvement probability, we conduct three steps: Abduction, Intervention, and Simulation [Pearl, 2009]: Specifically, to compute a counterfactual using a SCM, we first use the observation $x$ to abduct the unobserved causal influences, that is, we infer the posterior $P(U \mid X = x)$. Then, we implement the action in the SCM by replacing the affected structural equations. Last, we sample from the abducted noise $P(U \mid X = x)$ and generate the new outcomes using the updated structural equations. Based on the obtained sample from the *counterfactual distribution* we estimate the probability of a favorable outcome. For ICR, that is the probability of the favorable label $L = 1$, and for CR that is the probability of the favorable prediction $\widehat{L} = 1$.

In our experiments we focus on settings with invertible structural equations. That is, given a full observation $x, y$, we could compute the unique corresponding noise value $u$. To perform the abduction in a setting where $Y$ is not observed, a direct computation of the noise variables is not possible. Specifically, $U_Y$ and $U_E$ cannot be determined. However, for each possible $u_Y$, there is exactly one possible $u_E$; more formally there exists a function $u_E = g(u_Y)$. And one can show that as a result the abducted distribution must be proportional to $P(U_Y = u_y)P(U_E = g(u_Y))$. To sample from this distribution, we employ rejection sampling.

**Lemma F.7 (Abduction with invertible structural equations).** *It holds that*

$$p(u_Y \mid x) = \frac{P(U_E = f_E^{-1}(x_S, x_C, f_Y(x_C, u_Y), x_E))p(u_Y)}{p(x_E \mid x_C, x_S)}.$$

*Proof.*

$$
\begin{aligned}
p(u_Y \mid x) &= p(u_Y | x_C, x_S, x_E) \\
&= \frac{p(x_C, x_S, x_E, u_Y)}{p(x_C, x_S, x_E)} \\
&= \frac{\int_{\mathcal{U}_E} p(x_C, x_S, x_E, u_Y, u_E) \, \mathrm{d}u_E}{p(x_C, x_S, x_E)} \\
&= \frac{\int_{\mathcal{U}_E} p(x_E | x_S, x_C, u_Y, u_E) p(x_S, x_C) p(u_Y, u_E) \, \mathrm{d}u_E}{p(x_E \mid x_C, x_S) p(x_C, x_S)} \\
&= \frac{\int_{\mathcal{U}_E} p(x_E \mid x_S, x_C, u_Y, u_E) p(u_Y) p(u_E) \, \mathrm{d}u_E}{p(x_E \mid x_C, x_S)} \\
&= \frac{\int_{\mathcal{U}_E} \mathbb{1}[x_E = f_E(x_S, x_C, f_Y(x_C, u_Y), u_E)] p(u_Y) p(u_E) \, \mathrm{d}u_E}{p(x_E \mid x_C, x_S)} \\
&= \frac{\int_{\mathcal{U}_E} \mathbb{1}[u_E = f_E^{-1}(x_S, x_C, f_Y(x_C, u_Y), x_E)] p(u_Y) p(u_E) \, \mathrm{d}u_E}{p(x_E \mid x_C, x_S)} \\
&= \frac{P(U_E = f_E^{-1}(x_S, x_C, f_Y(x_C, u_Y), x_E)) p(u_Y)}{p(x_E \mid x_C, x_S)}.
\end{aligned}
$$

$\square$

**Estimating subpopulation-based outcomes** When no SCM is available, the causal recourse methods resort to a causal graph. The causal graph does not allow to abduct unobserved causal influences and therefore does not allow the computation of individualized effects. However, causal graphs allow to describe interventional distributions: Therefore, the joint distribution is factorized

into the conditional distributions of each node given its direct causal parents $P(X) = \prod_j P(X_j \mid X_{\mathrm{pa}(j)})$, and the conditional distributions for the intervened upon nodes $j \in I_a$ are replaced with $P(X_j) = \mathbb{1}[X_j = \theta_j]$. To obtain the interventional distribution for the subpopulation instead of the whole population, we additionally intervene on the nondescendants $X_{\mathrm{nd}}(I_a)$ to hold those values fixed [König et al., 2023]. In our experiments, instead of learning the conditional distributions, we obtain the ground-truth conditional distributions from the SCM: To sample from the interventional distribution, we sample from the intervened-upon SCM. In this sample, we again compute the probabilities of the favorable outcomes.

**Translating CEs into actions**    While CR and ICR recommend actions in the form of an intervention $\mathrm{do}(X_{I_a} = \theta_{I_a})$ on a subset of the variables $I_a$, CE only suggests a new observation $x'$. To translate this to a causal action, we intervene on *all* variables, and set them to the values specified by $x$. That is, $\mathrm{do}(X = x')$. In this setting, there is no uncertainty about the post-recourse observation and as a result the acceptance probability can directly be evaluated to zero or one.

## F.5   Optimization

To solve the optimization problems imposed by each of the recourse methods, we employ evolutionary algorithms (as proposed by Dandl et al. [2020]) and rely on the python package `deap` [Fortin et al., 2012]. Evolutionary algorithms can naturally deal with both continuous and categorical data. To optimize a goal, they randomly draw suggestions (individuals). This sample (population) is then modified and filtered over many rounds (generations).

In our setting, each individual consists of two values per variable: A binary indicator $\alpha_j$ that represents whether the variable $X_j$ shall be changed, and a float $\beta_j$ that represents the respective new value.

When initializing the population, we randomly draw $\alpha_j \sim \mathrm{Bin}(1, 0.5)$. If the variable is continuous, $\beta_j$ is sampled uniformly from the range $\mathrm{Unif}(b_l, b_u)$, where the lower and upper bound $b_l$ and $b_u$ are determined empirically from a very large sample from the distribution. If the variable is categorical, $\beta_j$ is drawn from the uniform distribution over the empirical support. To cross two individuals, we switch the $\alpha$ and $\beta$ values for each variable with $0.5$ probability. To mutate an individual, we modify each entry with $0.2$ probability, where the $\alpha_j$ values are simply flipped, and the $\beta_j$ values are either resampled if they are categorical or modified using a Gaussian kernel if they are continuous. We chose the population size 25, the number of generations 25, the crossing probability 0.5, and the mutation probability 0.5.

To select the individuals that make it to the next generation, we rely on the following fitness function:

$$cost(x, a) + \lambda(t_r - p^{\mathrm{success}}(x, a))$$

where $p^{\mathrm{success}}$ is the probability of a favorable outcome for the given recourse methods (as defined in Section 3 and Appendix B), and $\lambda = 10^4$.

## F.6   Computational resources

We ran the experiments on a MacBook Pro with M3 Pro Chip and a cluster with Intel Xeon Gold processors with 16 cores and 2.9GHz. The main compute is required for sampling from the post-recourse distributions, which takes roughly one hour on one core for one setting and one method, which amounts to a total of 30 hours per run and 300 hours for all experiments. We note that the runs were parallelized over the 16 cores. The post-processing of the results was done on the M3 Pro Chip and requires negligible computational effort.

