# OpenReview forum: "Performative Validity of Recourse Explanations"
_NeurIPS.cc/2025/Conference — NeurIPS 2025 poster_

### Official Review · Reviewer_jQ9B · 2025-06-25

**Clarity:** 3
**Significance:** 2
**Originality:** 3
**Rating:** 4
**Confidence:** 3

**Summary:**

The authors introduce the terms performative effect of recourse explanations and performative validity.
They then investigate whether recourse explanations remain valid under their performative effects. They theoretically prove that recourse becomes performatively invalid only if the implemented action
carries predictive information about the post-recourse outcome.
Lastly, they empirically study the performative validity of three recourse methods: counterfactual explanations, causal recourse, and improvement-focused causal recourse.

**Questions:**

- While certain features, such as a master's degree, may seem more costly to manipulate compared to others, like GitHub activity, this does not guarantee that acting on them truly reflects genuine qualification.  In the current landscape of generative AI tools,
(e.g., ChatGPT and Gemini, among others), and cheating rings, among other shortcuts, even ostensibly robust signals (e.g., an academic degree) are manipulable (effect variables).  The feature-level recourse methods (whether CE, CR, or ICR) that suggest how much a user should change each feature ("Translating CEs into Actions" G4 in the appendix) are vulnerable to performative invalidity in practice.
Ultimately, decision-makers only see the (new) revealed features, not the underlying process by which users reach there.

- I presume there are cases where the variable "a" is an effect variable in model 1 but is no longer an effect variable in model 2. Conversely, the variable "a" may not be an effect variable in model 1 but becomes an effect variable in model 2. For example, a model in the year 2012 that used Github activity versus a model in 2025 that used Github activity. How robust are the structural causal model generation processes to these sorts of changes, and how does that affect the performative validity?

- I need some clarification. If "neither the decision model nor the recourse algorithm is influenced by information about GitHub activity to arrive at their outputs," why is GitHub activity considered a feature in the model training? Why is it appearing in the SCM?

- Experimental details are sparse, and lots of key information is missing.
   - Could the authors include more details on the datasets (size, features, cleaning, etc.) and the generated SCMs?
   - From the experimental setting (G.2), the authors say they use logistic regression and decision-tree models. However, it's unclear how accurate they are on the datasets, how the models change after retraining, and so on. It's also unclear to me which model they use on which dataset and when.
   - How did the authors compute L^p in the ICR case?


- From the experimental setup, the authors mention that all settings include one cause and one effect variable. I am unsure how scalable the methods are.

- Since the authors' work is more aligned with the robust CFE generation approaches, I am curious about the empirical comparative analysis between the robust CFE generation methods (https://www.ijcai.org/proceedings/2024/0894.pdf) and the authors' proposed perspective.

- Since theoretical findings show that performative validity depends on the recourse method, would action-based, not feature-level, recourse methods, as those authors use, and would sequence-based recourse methods, and not the one-shot recourse methods authors use, do as well as the ICR at a computationally cheaper rate?

**Ethical Concerns:**

["NO or VERY MINOR ethics concerns only"]

**Final Justification:**

The authors provided detailed responses to the questions I had

**Limitations:**

No, there is no limitations section in the main paper or appendix.
It would be good to comprehensively address both the theoretical and empirical limitations of their work.
Please see the weakness section and questions.

**Paper Formatting Concerns:**

To the best of my knowledge, the paper is well formatted according to the formatting instructions.

**Quality:**

2

**Strengths And Weaknesses:**

**Strengths**
- The authors introduce a new perspective on recourse generation and connect the performative prediction to the CFE/recourse generation.
- The problem is interesting and offers useful insights to the community.
- The theory is well-written and easy to comprehend. Also, Figure 2 is intuitive.

**Weaknesses**

- While certain features, such as a master's degree, may seem more costly to manipulate compared to others, like GitHub activity, this does not guarantee that acting on them truly reflects genuine qualification.  In the current landscape of generative AI tools,
(e.g., ChatGPT and Gemini, among others), and cheating rings, among other shortcuts, even ostensibly robust signals (e.g., an academic degree) are manipulable (effect variables).  The feature-level recourse methods (whether CE, CR, or ICR) that suggest how much a user should change each feature ("Translating CEs into Actions" G4 in the appendix) are vulnerable to performative invalidity in practice.
Ultimately, decision-makers only see the (new) revealed features, not the underlying process by which users reach there.

- I presume there are cases where the variable "a" is an effect variable in model 1 but is no longer an effect variable in model 2. Conversely, the variable "a" may not be an effect variable in model 1 but becomes an effect variable in model 2. For example, a model in the year 2012 that used Github activity versus a model in 2025 that used Github activity. How robust are the structural causal model generation processes to these sorts of changes, and how does that affect the performative validity?

- I need some clarification. If "neither the decision model nor the recourse algorithm is influenced by information about GitHub activity to arrive at their outputs," why is GitHub activity considered a feature in the model training? Why is it appearing in the SCM?

- Experimental details are sparse, and lots of key information is missing.
   - Could the authors include more details on the datasets (size, features, cleaning, etc.) and the generated SCMs?
   - From the experimental setting (G.2), the authors say they use logistic regression and decision-tree models. However, it's unclear how accurate they are on the datasets, how the models change after retraining, and so on. It's also unclear to me which model they use on which dataset and when.
   - How did the authors compute L^p in the ICR case?


- From the experimental setup, the authors mention that all settings include one cause and one effect variable. I am unsure how scalable the methods are.

- Since the authors' work is more aligned with the robust CFE generation approaches, I am curious about the empirical comparative analysis between the robust CFE generation methods (https://www.ijcai.org/proceedings/2024/0894.pdf) and the authors' proposed perspective.

- Since theoretical findings show that performative validity depends on the recourse method, would action-based, not feature-level, recourse methods, as those authors use, and would sequence-based recourse methods, and not the one-shot recourse methods authors use, do as well as the ICR at a computationally cheaper rate?


**Miscellaneous**

- When characterizing the invalidity of the recourse, the authors frame the problem from the user's perspective. That is, the issue seems to mainly affect users (unless there is a computed true positive minus false positive rate or true negative minus false negative rate). Therefore, I am curious, what incentives do the decision-makers have to choose an expensive and difficult method like ICR (e.g., gathering causal knowledge is not trivial and is costly) over CE methods?
- Synthetically generated noise is a feasible way to test the model changes, but it does not sufficiently capture the complexities of change in data distributions and model drift.
- Authors look at invalidity from the perspective of the recourse not making the user qualified anymore. How about if h(x')=1 but f(x')=0? What are the implications from the casual perspective and long-term effects on both the user and the decision-maker?
- I am curious about the fairness implications of the suggested setup.


**Minor**
- Typos in the paper, e.g., decison instead of decision

---

> ### Author Rebuttal · Authors · 2025-07-31
>
> We are glad to hear that the reviewer finds our paper to be well-written and our insights to be original and useful to the community.
>
> *W1: [Is an academic degree really a robust signal (in contrast to GitHub activity)?]*
>
> Indeed, with the advent of tools like ChatGPT, academic degrees may have lost some of their predictive power. However, people have always attempted to cheat their way into an academic degree, and universities have consistently sought to prevent such practices. As such, we would expect that the recourse recommendation "get an academic degree" does not fundamentally alter the relationship between the degree and skill. This is not the case for a comparatively easy-to-manipulate feature such as GitHub activity.
>
> *How robust are the structural causal model generation processes to [changes in the causal structure], and how does that affect the performative validity?*
>
> By definition, performative validity only concerns shifts resulting from recourse. For this purpose, we assume that all other factors remain constant. Of course, other shifts may occur in practice, and studying them together is an interesting avenue for future research.
>
> *If "neither the decision model nor the recourse algorithm is influenced by information about GitHub activity to arrive at their outputs," why is GitHub activity considered a feature in the model training?*
>
> You cite from line 213 in the paper, where we argue that performative validity could be guaranteed if model and recourse method were to abstain from noncausal variables like GitHub activity. This may not be the case in practice: Effect variables can be highly predictive of the outcome, and abstaining from them would result in a loss of predictive accuracy. To make this clearer, think of a disease diagnosis setting: Although the symptoms of a disease are not causal for the disease (meaning that treating the symptoms does not make you healthy), they are still valuable for predicting whether somebody is ill or not!
>
> *Experimental details are sparse, and lots of key information is missing.*
>
> Details on the experiments are provided in Appendix G. The code was made available to the reviewers and will be made publicly available upon publication.
>
> *Could the authors include more details on the datasets (size, features, cleaning, etc.) and the generated SCMs? From the experimental setting (G.2), the authors say they use logistic regression and decision-tree models. However, it's unclear how accurate they are on the datasets, how the models change after retraining, and so on. It's also unclear to me which model they use on which dataset and when. How did the authors compute L^p in the ICR case?*
>
> The full specification of the SCMs is reported in Appendix G.1. The specification of the models is reported in G.2. Decision trees are used in the synthetic settings, and a logistic regression model for the real-world example (SATGPA). In all settings, $L^p$ is defined as $Y \geq \text{median}(Y)$ to ensure balanced classes.
>
> We updated G.2 to report test set accuracies (averaged over 10 runs) for the Original and Refitted models, both pre- (O and R) and post-recourse (OP and RP).
>
> | SCM      | Type     |    O |    R |   OP |   RP |
> |:---------|:---------|-----:|-----:|-----:|-----:|
> | LinAdd   | CE       | 0.91 | 0.9  | 0.51 | 0.89 |
> | LinAdd   | ind. ICR | 0.91 | 0.91 | 0.96 | 0.96 |
> | LinAdd   | sub. CR  | 0.91 | 0.9  | 0.53 | 0.89 |
> | LinAdd   | ind. CR  | 0.91 | 0.9  | 0.51 | 0.89 |
> | LinAdd   | sub. ICR | 0.91 | 0.91 | 0.96 | 0.96 |
> | LinMult  | sub. CR  | 0.87 | 0.87 | 0.5  | 0.85 |
> | LinMult  | ind. ICR | 0.87 | 0.87 | 0.94 | 0.94 |
> | LinMult  | sub. ICR | 0.87 | 0.87 | 0.94 | 0.94 |
> | LinMult  | CE       | 0.87 | 0.87 | 0.5  | 0.85 |
> | LinMult  | ind. CR  | 0.87 | 0.87 | 0.5  | 0.85 |
> | NlinAdd  | sub. ICR | 0.91 | 0.91 | 1    | 0.99 |
> | NlinAdd  | ind. ICR | 0.91 | 0.91 | 0.99 | 0.99 |
> | NlinAdd  | sub. CR  | 0.91 | 0.9  | 0.58 | 0.83 |
> | NlinAdd  | ind. CR  | 0.91 | 0.9  | 0.58 | 0.83 |
> | NlinAdd  | CE       | 0.91 | 0.89 | 0.58 | 0.83 |
> | NlinMult | sub. CR  | 1    | 1    | 0.84 | 0.83 |
> | NlinMult | sub. ICR | 1    | 1    | 1    | 1    |
> | NlinMult | ind. ICR | 1    | 1    | 1    | 1    |
> | NlinMult | CE       | 1    | 1    | 0.84 | 0.84 |
> | NlinMult | ind. CR  | 1    | 1    | 0.84 | 0.92 |
> | LinCubic | sub. CR  | 0.92 | 0.86 | 0.54 | 0.83 |
> | LinCubic | CE       | 0.92 | 0.86 | 0.54 | 0.83 |
> | LinCubic | ind. ICR | 0.92 | 0.92 | 0.95 | 0.95 |
> | LinCubic | ind. CR  | 0.92 | 0.86 | 0.55 | 0.83 |
> | LinCubic | sub. ICR | 0.92 | 0.92 | 0.96 | 0.96 |
> | GPA      | sub. ICR | 0.73 | 0.73 | 0.8  | 0.8  |
> | GPA      | CE       | 0.73 | 0.72 | 0.51 | 0.7  |
> | GPA      | ind. ICR | 0.73 | 0.73 | 0.81 | 0.81 |
> | GPA      | ind. CR  | 0.73 | 0.7  | 0.51 | 0.68 |
> | GPA      | sub. CR  | 0.73 | 0.72 | 0.51 | 0.69 |
>
>
> We note that predictive accuracy is not within the scope of the present paper. The model’s accuracy may drop even if the model and conditional distribution $P(Y|X)$ remain the same after recourse. The reason is that recourse may move to regions with higher uncertainty, as demonstrated by [1].
>
> [1] Fokkema, Hidde, Damien Garreau, and Tim van Erven. "The risks of recourse in binary classification." International Conference on Artificial Intelligence and Statistics. PMLR, 2024.
>
> *From the experimental setup, the authors mention that all settings include one cause and one effect variable. I am unsure how scalable the methods are.*
>
> We chose low-dimensional, discrete settings to enable an accurate empirical comparison of the pre- and post-recourse conditional distributions (Q1). Increasing the dimensionality would require significantly more samples from the post-recourse distribution, which are expensive to obtain since each sample requires solving the recourse optimization problems.
>
> To address your concern, we added another real-world setting with 11 features, based on the UCI “Default of credit card client” dataset. We chose causes and effects as suggested in [1]. We fit the SCM using random forests, assuming Gaussian noise. The results confirm our theory and previous experiment results: The acceptance rates (Q2) remain the same for ICR, but drop significantly for CR and CE.
>
> | SCM    | Type     |   Acc. Rate Diff. Mean |   Acc. Rate Diff. Std |
> |:-------|:---------|-----------------------:|----------------------:|
> | Credit | CE       |                  -0.47 |                  0.05 |
> | Credit | sub. CR  |                  -0.39 |                  0.04 |
> | Credit | ind. CR  |                  -0.49 |                  0.03 |
> | Credit | sub. ICR |                   0.01 |                  0.01 |
> | Credit | ind. ICR |                   0.01 |                  0.01 |
>
> [1] Chen, Yatong, Jialu Wang, and Yang Liu. "Learning to incentivize improvements from strategic agents." Transactions on Machine Learning Research (2023).
>
> *[...] would action-based, not feature-level, recourse methods, as those authors use, and would sequence-based recourse methods, and not the one-shot recourse methods authors use, do as well as the ICR at a computationally cheaper rate?*
>
> With CR and ICR, we include two action-based methods. They formalize actions as causal interventions. Only CE does not specify the action to be taken to achieve the desired feature values; therefore, we transform them into an action-based method by intervening on every feature (Karimi, FAccT 2021). Does this answer your question? If not, can you clarify what exactly you mean by "action-based" and “sequence-based” recourse?
>
> *When characterizing the invalidity of the recourse, the authors frame the problem from the user's perspective. [...]. Therefore, I am curious, what incentives do the decision-makers have to choose an expensive and difficult method like ICR (e.g., gathering causal knowledge is not trivial and is costly) over CE methods?*
>
> ICR is clearly more desirable for the decision maker. CE and CR are allowed to intervene on noncausal features and may thus instruct the applicant on how to deceive the predictor into believing they are qualified without actually making them qualified (acceptance without improvement). This conflicts with the decision maker's objective of making accurate decisions!
>
> *[...]. How about if h(x')=1 but f(x')=0? What are the implications from the casual perspective and long-term effects on both the user and the decision-maker?*
>
> With h(x’)=1 and f(x’)=0, do you mean that the original model would accept the individual post-recourse but the post-recourse label is negative (false positive)? This is precisely the case that we study! As a result, the post-recourse conditional probability of a positive outcome decreases, and recourse recommendations may be invalidated once the model is updated.
>
> *I am curious about the fairness implications of the suggested setup.*
>
> This is an interesting question for follow-up work. In particular, it is conceivable that performative invalidity affects different groups disproportionately (i.e., actions on noncausal variables are more likely to be suggested to one group than another), further highlighting the importance of performative validity.
>
> *[Are “robust CE” methods performatively valid?]*
>
> Robust variants of CE and CR are tailored to generic shifts; for instance, they may suggest moving not just onto but further past the decision boundary. However, they do not distinguish between interventions on causes and effects, and thus, in general, do not avoid interventions on noncausal effect variables. As such, we expect them to lead to performative invalidity. We conjecture that combining ICR with ideas from the literature on robust recourse will lead to recourse recommendations that are more expensive but not only performatively valid but also more robust to other types of shifts.
>
> **We hope our clarifications were helpful in addressing your questions. If they were, we’d greatly appreciate it if you would consider updating your score.**

---

> ### Comment · Reviewer_jQ9B · 2025-08-04
>
> Thank you for providing the updated version of G.2, adding the credit card dataset, and for all the detailed responses to the  questions I had. I’ve adjusted the score

---

### Official Review · Reviewer_c6Gq · 2025-06-27

**Clarity:** 4
**Significance:** 3
**Originality:** 4
**Rating:** 5
**Confidence:** 4

**Summary:**

This paper investigates performative effects, a phenomenon where widespread adoption of recourse recommendations shifts the data distribution, potentially rendering the recommendations ineffective after the model is retrained. To address this, the authors introduce the notion of *performative validity* and formally characterize the conditions under which it can be preserved. They show that performative validity fails when recourse actions are either influenced by effect variables or directly intervene on them. Empirical evaluations support the theoretical findings, confirming that avoiding effect variables helps maintain performative validity.

**Questions:**

Are there any reliable sources that show how often performative effects occur in the real world? I'm ready to raise the Significance score if they're a frequent issue.

**Ethical Concerns:**

["NO or VERY MINOR ethics concerns only"]

**Final Justification:**

The rebuttal improved my significance and confidence score. The main weaknesses have been addressed, and this paper is worthy of acceptance.

**Limitations:**

yes

**Quality:**

3

**Strengths And Weaknesses:**

- Strengths
  - Originality: The paper’s focus on performative effects and its formalization of performative validity is novel. This perspective has received limited attention in prior work, and the study is likely to inspire follow-up research in this emerging area.
  - Quality: The finding that *improvement-focused causal recourse (ICR)* consistently maintains performative validity across diverse data-generating processes—such as resampling and additive noise—is theoretically and empirically compelling. To the best of my knowledge, this is the first paper to establish the superiority of ICR over CE and CR from the perspective of performative validity.
  - Clarity: The paper explains related work, prerequisite concepts, and proposed methods with clarity and depth. Examples (e.g., Figure 1) and causal diagrams (e.g., Figure 2) effectively illustrate key ideas. Furthermore, thoughtful presentation choices, including text color and layout, enhance readability. The citations also appropriately cover the foundational work needed to situate this study’s contribution.
- Weaknesses
  - Significance: The study's practical impact could be better emphasized. While research on algorithmic recourse has gained substantial attention in academic circles, its societal deployment remains limited. Moreover, it is unclear which real-world domains performative effects occur in and how frequently. Some discussion or empirical justification on this point—possibly based on the experimental results in Q1—would be valuable. Although the GPA prediction task offers a semi-realistic use case, assumptions such as the frequency of model retraining may limit its generalizability.
  - Practical scope: The reliance on causal knowledge narrows the proposed approach's applicability. As the authors acknowledge in Section 6, extending the framework to settings with incomplete or imperfect causal understanding is an essential and promising direction for future work.

---

> ### Author Rebuttal · Authors · 2025-07-31
>
> We are glad to hear that you find our work to be original, of high quality, compelling, and clear.
>
> ### W1: Emphasis on the practical impact
>
> *The study's practical impact could be better emphasized. While research on algorithmic recourse has gained substantial attention in academic circles, its societal deployment remains limited. Moreover, it is unclear which real-world domains performative effects occur in and how frequently. Some discussion or empirical justification on this point—possibly based on the experimental results in Q1—would be valuable. Although the GPA prediction task offers a semi-realistic use case, assumptions such as the frequency of model retraining may limit its generalizability.*
>
> ML is increasingly being used for high-stakes decisions [1], and recourse has been identified as one of the central requirements towards restoring the agency of affected individuals [2]. Concretely, their application has been suggested in various contexts such as employment services [3], hiring marketplaces [4], health [5,12], academic performance [6,7,9,10], and credit risk [8,11].
>
> Recourse explanations provide concrete action recommendations to individuals. Thus, as soon as at least some individuals act upon them, they do have a performative effect on the underlying distribution. Our theory concerns the important open question of whether and under which assumptions the performativity of recourse affects the validity of recourse recommendations. Even if we had found that recourse is always performatively valid, that would be a valuable new insight!
>
> Our theoretical and experimental results show that performative validity may be violated when decision model or recourse algorithm rely on noncausal variables. Noncausal variables are parsimonous and often attractive variables for prediction since they can be highly predictive and are often more accessible than the respective causes. Examples include disease symptoms in the context of medical diagnosis/allocation of health resources [5,12], the CV word count in the context of hiring [4], or how long ago the customer last switched their bank in the context of credit approval. As follows, we illustrate the case of credit risk scores in more detail.
>
> Credit risk scores (such as FICO or SCHUFA) use predictive models to classify the solvency of consumers. They act as gatekeepers in important contexts such as loan application or on the housing market. In recent years, more and more companies, including FICO and SCHUFA, offer so-called credit risk simulators that give actionable recommendations on how to improve risk scores.
> One of the most important features for FICO is credit utilization—the ratio of a person's current credit card balance to their total available credit (see e.g. the American Express MyCredit Guide). This feature serves as a proxy for financial discipline and is strongly associated with default risk. However, most credit scoring models only observe credit utilization at a single point in time—typically the time of application. This means that as long as an applicant’s credit utilization is low at that specific moment, it does not matter whether they usually max out their credit limits.
> If recourse explanations suggest that applicants lower their credit utilization just before applying, the predictive value of this feature may diminish, since the feature may no longer reliably indicate financial discipline or default risk. As a result, recourse users who follow these suggestions may not be rewarded in future decision rounds, as credit models will be retrained on behavior that no longer correlates with risk -- rendering the recourse ineffective or even counterproductive.
>
> [1] Mehrabi, Ninareh, et al. "A survey on bias and fairness in machine learning." ACM computing surveys (CSUR) 54.6 (2021): 1-35.
>
> [2] Venkatasubramanian, Suresh, and Mark Alfano. "The philosophical basis of algorithmic recourse." Proceedings of the 2020 conference on fairness, accountability, and transparency. 2020.
>
> [3] Raphael Mazzine, Sofie Goethals, Dieter Brughmans, and David Martens. Counterfactual
> Explanations for Employment Services. International workshop on AI for Human Resources728
> and Public Employment Services, 2021.
>
> [4] Nemirovsky, Daniel, et al. "Providing actionable feedback in hiring marketplaces using generative adversarial networks." Proceedings of the 14th ACM International Conference on Web Search and Data Mining. 2021.
>
> [5] Ong, Ming Lun, Anthony Li, and Mehul Motani. "Explainable and Actionable Machine Learning Models for Electronic Health Record Data." International Conference on Biomedical Engineering. Cham: Springer International Publishing, 2019.
>
> [6] Tsiakmaki, Maria, and Omiros Ragos. "A case study of interpretable counterfactual explanations for the task of predicting student academic performance." 2021 25th International Conference on Circuits, Systems, Communications and Computers (CSCC). IEEE, 2021.
>
> [7] Wijekoon, Anjana, et al. "Counterfactual Explanations for Student Outcome Prediction with Moodle Footprints." SICSA XAI. 2021.
>
> [8] Davis, Randall, et al. "Explainable machine learning models of consumer credit risk." Available at SSRN 4006840 (2022).
>
> [9] Ramaswami, Gomathy, Teo Susnjak, and Anuradha Mathrani. "Supporting students’ academic performance using explainable machine learning with automated prescriptive analytics." Big Data and Cognitive Computing 6.4 (2022): 105.
>
> [10] Smith, Bevan I., Charles Chimedza, and Jacoba H. Bührmann. "Individualized help for at-risk students using model-agnostic and counterfactual explanations." Education and Information Technologies 27.2 (2022): 1539-1558.
>
> [11] Wang, Zijie J., et al. "Gam coach: Towards interactive and user-centered algorithmic recourse." Proceedings of the 2023 CHI Conference on Human Factors in Computing Systems. 2023.
>
> [11] Lacerda, Anisio, et al. "Algorithmic Recourse in Mental Healthcare." 2023 International Joint Conference on Neural Networks (IJCNN). IEEE, 2023.
>
>
>
> ### W2: Causal knowledge
>
> *The reliance on causal knowledge narrows the proposed approach's applicability. As the authors acknowledge in Section 6, extending the framework to settings with incomplete or imperfect causal understanding is an essential and promising direction for future work.*
>
> Indeed, the requirement of causal knowledge is one of the key factors that limits the application of methods like CR and ICR -- they require at least knowledge of the causal graph. Causal knowledge is an inevitable requirement for recourse, since it is an inherently causal problem.
>
> In contrast, for our main results (4.1 & 4.2), you only need to know whether causal sufficiency holds and whether non-causal variables are involved, which is a weaker requirement than knowing the full causal graph, making them more practically relevant. 4.4 and 4.6 can best be assessed given the complete SCM; Whether the SCM is always required is an interesting question for future work.
>
> **We hope our clarifications were helpful in addressing your questions. If they were, we’d greatly appreciate it if you would consider updating your score.**

---

> > ### Comment · Reviewer_c6Gq · 2025-08-04
> >
> > Thank you for providing so many use cases. My confidence in my evaluation has increased.

---

### Official Review · Reviewer_gDwE · 2025-06-29

**Clarity:** 1
**Significance:** 2
**Originality:** 2
**Rating:** 4
**Confidence:** 4

**Summary:**

The goal of this paper is to address the problem of reliable algorithmic recourse (counterfactual explanations) specifically taking into account how recourses can change the distribution which in turn changes the models. Specifically: when many applicants act according to the recourse recommendations, their collective behavior may shift the data distribution >> shifts the new model trained on the updated data >> changes the model boundary >> makes the recourses invalid (since they are typically the closest points on the decision boundary). The paper investigates this crucial problem through a causal lens, showing that recourses may become invalid if they are influenced by  non-causal variables, and hence, advocates for recourses that specifically focus on causal variables.

**Questions:**

I have included several questions along with the Weaknesses.
It would be great to get clarifications for these questions.
With some more clarifications, this can be quite a powerful result.

**Ethical Concerns:**

["NO or VERY MINOR ethics concerns only"]

**Final Justification:**

I liked the core idea of the paper, though there were some technical inconsistencies and missing information/clarity in several places that I pointed out.
The authors have responded to my questions to the extent that they could in the rebuttal.
I maintain my positive rating since I generally agree with their responses, but the rebuttal has little details.
For a paper primarily making theory contributions, I would expect more clarity and toy examples in the presentation.

**Limitations:**

Some discussion is included.
Given the nature of this work, I would encourage more elaborate discussion on the physical meaning of their assumptions, and the role that they would play on the conclusions derived.

**Quality:**

3

**Strengths And Weaknesses:**

**Strengths:**
This is a very interesting perspective on the challenge of reliable algorithmic recourse. Prior works have looked into this problem, aiming to provide a robust algorithmic recourse (or, robust counterfactual explanations) that finds recourses that would remain valid even under model changes. This work takes a different perspective on this problem, bringing in several novel elements:
1. noting that data distributions change when people act upon the recourses, potentially leading to certain model changes (rather than a random model change)
2. bringing in a causal perspective into the problem, showing how recourses that alter non-causal (spurious) features/variables can change the data distribution in a way that new models might no longer depend on that feature, thus making the recourses invalid

Proposition 4.1 is quite a surprising result - particularly the if and ONLY if part of the result. Some more clarifications on the assumptions will be needed. After resolving some comments/questions/clarifications, this can be quite a powerful claim.

They have also included limited empirical results to validate the theoretical findings and included some thoughtful and clever toy examples (more clarifications needed though).

**Weaknesses: (includes questions)**

W1. The results are proven based on certain worldviews/assumptions on HOW the data distribution changes once a recourse is provided, what part of the population receives and acts upon the recourse, consequently leading to the data distribution change. It would be nice to highlight these assumptions/worldviews upfront (not just mathematically, but physically what they mean) for better interpretability of the mathematical results and the conclusions derived from them. This is quite important in interpreting the results because this is quite a surprising claim.

-- Q1: For the post-recourse distribution $P(X_p,Y_p)$, is it assumed that every point in the data distribution acts upon the recourse? Would this be a reasonable assumption in practice? What would be its physical meaning?
Does it have provision for:
 (i) only few individuals actually applying and receiving recourse and acting upon it? For instance, not all candidates might actually apply/interview/receive feedback. There might be several candidates who stay the same, and only a handful of them implement the recourse. (I am not sure if low $\alpha$ alone is sufficient to model this sparsity? Low $\alpha$ would still place a non-zero mass on *all* altered datapoints, but here only a handful of altered datapoints should have non-zero mass?)
(ii) entirely new candidates can get added to the pool who have not applied before.

-- Q2: The updated data distribution is assumed to be a mixture of $P(X,Y)$ and $P(X_p,Y_p)$ weighted by $\alpha$ and $1-\alpha$. Would this be a reasonable assumption in practice and what is its physical meaning?
-- Q3: What happens when $\alpha$ tends towards 0 or 1?

W2. Similarly, there are also implicit assumptions being made on how the new model is being trained once the data changes which need to be made explicit.
-- Q4: Are you assuming that the new model does not consider the data imbalance in the new data distribution? Depending on the $\alpha$, there might be severe data imbalance in the new data distribution. What if the new model is trained after fixing the data imbalance, e.g., by reweighting?

W3. A clear example will be very helpful. Though Examples 4.3 and 4.8 is provided, they lack a lot of details and have some notational typos (I also checked the details in Appendix E and F, but still have questions):

It says L is 1 when Y>0. Is L the true label or the predicted label?
Is L_hat/h a deterministic function of the features Xc and XE?
What is beta?
Could it be clarified what is the initial model h (perhaps provide a concrete model as an example, e.g., a linear classifier)? Then, for this model h and a chosen alpha, what is the hm? Then, for this example, could you show how does this statement mathematically play out: "Then, the examination score becomes less predictive of the patient’s risk."
After acting upon the recourse, are you assuming true label L will change if one acts upon Xc but not if one acts on Xe?

W4. There are typos in the proofs in the Appendix, and also some details are missing in several places. For instance, Y and L have been used interchangeably in places. Some writing errors too.

W5. While the mathematical results seem interesting, I am not fully clear on the final conclusions derived.

-- Q4: What happens if the recourses are about the causal feature, the population updates that causal feature, and why wouldn't those recourses get invalidated?

---

> ### Author Rebuttal · Authors · 2025-07-31
>
> We are glad to hear that you find our perspective “interesting” and “novel” in several ways, our claims to be “powerful”, and our examples to be “thoughtful and clever”. Furthermore, we appreciate the reviewer's constructive feedback.
>
> ### W1-2/Q1-4: Intuitive Explanations of the Assumptions
>
> *The results are proven based on certain worldviews/assumptions on HOW the data distribution changes once a recourse is provided, what part of the population receives and acts upon the recourse, consequently leading to the data distribution change. It would be nice to highlight these assumptions/worldviews upfront (not just mathematically, but physically what they mean) for better interpretability of the mathematical results and the conclusions derived from them. This is quite important in interpreting the results because this is quite a surprising claim.*
>
> Thank you for the helpful comment -- you raise a valid point. We commit to updating the paper to clarify the required assumptions and their underlying intuition.
>
> For our main results (4.1 and 4.2), we assume that (1) accepted individuals stay the same post-recourse, (2) the underlying causal mechanisms are the same post-recourse, and (3) the post-recourse distribution stems from causal interventions $A(x)$, which are a function of the applicants’ pre-recourse observations. Intuitively, (1) & (2) mean that there is no shift except for the one caused by recourse, and (3) -- although more general -- holds whenever all rejected individuals carry out the recommended recourse action. For results/experiments concerning specific recourse methods, we furthermore assume that $A(x)$ is given by the respective recourse method.
>
> Assumption (1) is captured in (Eq. 1). Apart from that, (Eq. 1) is very general: The applicant distribution $P(L, X)$, the post-recourse (or reapplicant) distribution $P(L^p, X^p|^L(X)=1)$, and $\alpha$ can be chosen freely. Assumptions (2) and (3) are only introduced later (in Section 3.2) to keep the definition of performative validity (Definition 3.2) as general as possible, such that it can serve as the starting point of follow-up work that relies on a different set of assumptions.
>
> Our assumptions may not hold in practice, for example, additional external distribution shifts may occur, and whether one acts on a recommendation may depend on external factors. However, we highlight that in our analysis, the assumptions serve the purpose of isolating and studying the shifts *caused by specific recourse methods*. We make no claims about the robustness of recourse to other types of distribution shifts!
>
> **Addressing your concrete questions:**
>
> (Q1)(i) Yes, the applicant distribution $P(X, L)$ can be a subpopulation of some larger population. The recourse action must be expressible with $A(x)$, and $A(x)$ can also map to $do(\emptyset)$ (do nothing). It is safe to assume that an i.i.d. sample of the rejected applicants implements the recourse recommendation and reapplies. (ii) Yes, as long as the new applicants are an i.i.d. sample from the original applicant distribution $P(X, L)$.
>
> (Q2) $\alpha$ allows for adjusting the weight given to the original applicant distribution and the post-recourse reapplicant distribution. As discussed above, (Eq. 1) is very general.
>
> (Q3) The larger $\alpha$, the smaller the impact of post-recourse reapplications in the mixture. The choice of $\alpha$ does not matter for our main results, as they concern whether a shift occurs, rather than the magnitude of the shift.
>
> (W2/Q4) We assume the model is designed to predict optimally in the mixture (with the respective $\alpha$), and thus no balancing is employed in our experiments. We note that the choice of alpha does not matter for our main results (see Q3).
>
> ### W3 and questions
>
> *A clear example will be very helpful. Though Examples 4.3 and 4.8 is provided, they lack a lot of details and have some notational typos (I also checked the details in Appendix E and F, but still have questions).*
>
> Thank you for the valuable reading impression. We commit to updating the description and proof to make the Example easier to parse.
>
> *It says L is 1 when Y>0. Is L the true label or the predicted label?*
>
> In line with the rest of the paper, $L$ is the true label. In Example 4.3, $L$ is a binary version of $Y$ that reflects whether a patient is qualified to spend the day outside, and $0$ is the qualification threshold.
>
> *Is $\hat{L}$/$h$ a deterministic function of the features Xc and XE?*
>
> Yes, $\hat{L}$ is the binary predictor, and thus a function of the features.
>
> *What is beta? Could it be clarified what is the initial model h is (perhaps provide a concrete model as an example, e.g., a linear classifier)?*
>
>
> In Example 4.8 the original model is linear and given by $(X_C + X_E) / \sqrt(2) >=0$. Why? The model is based on the conditional $h(x) = P(L=1 |X=x)=\Phi(X_C + X_C) / \sqrt{2})$, where $\Phi$ is the cumulative distribution function of a standard normal distribution, and the model's decision threshold $0.5$.
>
> *Then, for this model h and a chosen alpha, what is the hm?*
>
> The retrained model $h^m$ is a mixture between $h^p$ and $h$, depending on $\alpha \in (0,1)$. To derive the post-recourse model, we leverage Proposition F.1 which enables us to describe the post-recourse conditional distribution as a mixture between the original conditional distribution and the conditional distribution where the association between target and effects was broken. Specifically, $h^p(x)= P(L^p=1 |X^p=x) = (1 - \beta)P(L=1 | X =x) + \beta * P(L = 1 | X_C =x_C)$. It can again be shown that $P(L=1 |X_C=x_c)  = \Phi(x_C)$. The parameter $\beta$ is the proportion of post-recourse individuals that intervened on the effect, and in this Example assumed to be $>0$.
>
> *Then, for this example, could you show how does this statement mathematically play out: "Then, the examination score becomes less predictive of the patient’s risk."*
>
> Since $h^m$ is a mixture between $P(L=1|X=x)$ and $P(L=1|X_C=x_C)$, and since the second term is independent of the effects, the effects, in this case the examination scores, become less predictive post-recourse/in the refit.
>
> *After acting upon the recourse, are you assuming true label L will change if one acts upon Xc but not if one acts on Xe?*
>
> The reason the true label $L$ does not change when intervening on $X_E$, but $L$ does change when intervening on $X_C$ follows from the fact that $X_C$ is assumed to be a cause of $L$ and $X_E$ an effect of $L$.
>
> ### W5/Q5
>
> *While the mathematical results seem interesting, I am not fully clear on the final conclusions derived. What happens if the recourses are about the causal feature, the population updates that causal feature, and why wouldn't those recourses get invalidated?*
>
> Although interventions on causes may shift the distribution, they leave the relationship between features and target unaffected. In other words, while $P(X)$ may shift, $P(Y|X)$ stays the same. To make this more intuitive, think of a setting where an applicant’s Covid risk is used to determine whether somebody is allowed to enter a building. Two features are used to predict the covid risk: one cause, the vaccination status, and one effect, the body temperature. Taking antipyretics to get rid of fever (intervening on the symptom/effect) only makes someone appear to have a low covid risk without actually affecting the covid risk, thereby breaking the association between body temperature and covid risk.
> In contrast, getting a vaccination (intervening on the cause) does indeed decrease the covid risk and the statistical relationship between vaccination and covid risk is preserved.
>
> **We hope our clarifications were helpful in addressing your questions. If they were, we’d greatly appreciate it if you would consider updating your score.**

---

> > ### Comment · Reviewer_gDwE · 2025-08-05
> > **Read the rebuttal**
> >
> > Thank you for your detailed response. These details and toy examples should be included in any future version for improved clarity. I maintain my positive support.

---

> > ### Comment · Reviewer_rdGh · 2025-08-06
> >
> > - I thank the authors for their discussion of the relevancy of risks related to algorithmic recourse, and I particularly appreciate the examples given, in particular  credit scoring that may be one of the main important use case, and on the most documented . Why the authors did not use this example and work it out through the paper?
> >
> > - Is assumption 4.6 well satisfied in the examples? Is it possible to give the expression for linear Gaussian of the functions as an illustration of the assumption.
> >
> > I thank the authors for their clarification for my other questions.

---

> ### Author Response · Authors · 2025-08-07
>
> *I thank the authors for their discussion of the relevancy of risks related to algorithmic recourse, and I particularly appreciate the examples given, in particular credit scoring that may be one of the main important use case, and on the most documented . Why the authors did not use this example and work it out through the paper?*
>
> We are glad to hear that you like the examples and the credit scoring example. We initially thought that the hiring scenario was more intuitive, but we commit to updating the paper to reflect your feedback.
>
> *Is assumption 4.6 well satisfied in the examples? Is it possible to give the expression for linear Gaussian of the functions as an illustration of the assumption.*
>
> In general, we would not recommend making Assumption 4.6 without careful evaluation of the concrete setting. Since we have proven that assumptions such as 4.4 or 4.6 are necessary to guarantee performative validity, we recommend carefully studying the shift induced by recourse instead, using either our theoretical results or a simulation study.
>
> Assumption 4.6 is indeed satisfied in the SATGPA example, where the SCM is linear with additive Gaussian noise. That means that every variable can be written as
> $$X_j := U_j + \sum_{i \in Pa(j)} \beta_i X_i$$
>  where $U_j \sim N(\mu, \sigma)$ and $Pa(j)$ are the causal parents (direct causes) of the variable. To see this, we fitted linear models to predict each variable from its causes and diagnosed the residuals. We found the residuals to follow a Gaussian distribution both by visual inspection and by comparing the empirical CDF with the CDF of the corresponding Gaussian fit.
>
> We hope that we have addressed all your concerns, and thank the reviewer for their continued positive evaluation.

---

### Official Review · Reviewer_rdGh · 2025-07-02

**Clarity:** 3
**Significance:** 3
**Originality:** 3
**Rating:** 4
**Confidence:** 1

**Summary:**

This paper investigates the performative aspect of algorithmic recourse: when we provide an individual with a “recourse action” this very action may shift the model’s future behavior, invalidating the effectiveness of the recourse when the model is retrained. The authors  formalize the performative validity, a criteria to quantify when recourse remains valid under distributional shifts induced by the recourse itself, i.e. when users take a suggestion from a system and improve themselves, then the model retrained on new data points will not reject the users that would originally be accepted. The framework used are Structural Causal Models and they show that two mechanisms can induce such invalidity: actions that are influenced by effect variables and actions that intervene on effect variables, Finally,  the concepts are used to show that Counterfactual Explanations and Causal Recourse can lead to invalidity. While the Improvement-Focused Causal Recourse can satisfy this property.
The Formal Condition for Validity is that the recommended action $A$ is conditionally independent of the true post-recourse label $L^p$ given the post-recourse features $X^p$. This reduces the problem to a statistical independence condition: $A\sqcup L^{p}|X^{p}$. The theorems provide a clear, graph-based diagnostic for when a recourse method is at risk of invalidating itself.
Some experiments on SCM with various structural model show that ICR preserves the validity, while it is not the case for CE and CR.

**Questions:**

- can you give more practical / real life examples
- can we show mathematically that CE and CRa are not valid ?
- Do we need to know the causal graphs in advance to apply the results proposed in the paper? This seems very difficult in real world. I am also wandering what would be the difficulties if we don't assume assumption 4.4 and 4.6? }

**Ethical Concerns:**

["NO or VERY MINOR ethics concerns only"]

**Limitations:**

yes

**Quality:**

3

**Strengths And Weaknesses:**

Strengths and weaknesses.

The paper is well written with a good balance between necessary and exhaustive information (such as discussion of the state of the art, or proofs) between main paper and supplemental.
- Various examples are presented and discussed to illustrate the concept and properties of interests.
- the main theorems are clearly stated and well motivated and explained, and we can apply the methodology.


Weaknesses:
- The impact of retraining the model is not discussed on experiments or theoretically. In the paper, it is not clear what is the real impact, and it seems very useful to link the mixture model (1) (and alpha), with the number of rejected recourses or previously accepted observations. Could you clarify? What is the appropriate concept to anticipate the shift / amplitude of the drop in performance of the recourses.
- Assumption 4.6 seems to be strong : is it possible de relax it? Do we have equivalence with this assumption in theorem 4.7.
How can we use it in practice?
- I wonder about the significance or interests about this work: although I found the scientific question interesting, I am not fully convinced by the initial examples given in HR, and I feel thai it is a good illustration but not a real problem in practice. Do you have any alternatives?

---

> ### Author Rebuttal · Authors · 2025-07-31
>
> We thank the reviewer for the feedback. We are glad that you find the paper well-written, the examples helpful, and the results clear, well-motivated, and applicable.
>
> ### Q1/W3: Real-world Examples
>
> *I wonder about the significance or interests about this work: although I found the scientific question interesting, I am not fully convinced by the initial examples given in HR, and I feel thai it is a good illustration but not a real problem in practice. Do you have any alternatives? And: Can you give more practical / real life examples*
>
> Algorithmic recourse has been suggested in various contexts such as employment services [1], hiring marketplaces [2], health [3,10], academic performance [4,5,7,8], and credit risk [6,9]. Performativity is relevant for any of these examples where ML is applied to individuals, see [10] and references therein. Let us use credit scoring as an additional example to illustrate the performative effects of recourse which is used as a motivating example across the literature on performative prediction and strategic classification. Also, [11] discusses practical examples of performativity in financial markets.
>
> Credit risk scores (such as FICO or SCHUFA) use predictive models to classify the solvency of consumers. They act as gatekeepers in contexts such as loan applications or the housing market. In recent years, an increasing number of companies, including FICO and SCHUFA, have offered so-called credit risk simulators that provide actionable recommendations on how to improve risk scores.
> One of the most important features for FICO is credit utilization—the ratio of a person's current credit card balance to their total available credit (see e.g., the American Express MyCredit Guide). This feature serves as a proxy for financial discipline and is strongly associated with default risk. However, most credit scoring models only observe credit utilization at a single point in time—typically the time of application. This means that as long as an applicant’s credit utilization is low at that specific moment, it does not matter whether they usually max out their credit limits.
> If recourse explanations suggest that applicants lower their credit utilization before applying, then the predictive value of this feature diminishes. In such cases, it is conceivable that the feature no longer reliably indicates financial discipline or default risk. As a result, recourse users who follow these suggestions may not be rewarded in future decision rounds, as credit models will be retrained on behavior that no longer correlates with risk—rendering the recourse ineffective or even counterproductive.
>
>
> [1] Raphael Mazzine, Sofie Goethals, Dieter Brughmans, and David Martens. Counterfactual
> Explanations for Employment Services. International workshop on AI for Human Resources728
> and Public Employment Services, 2021.
>
> [2] Nemirovsky, Daniel, et al. "Providing actionable feedback in hiring marketplaces using generative adversarial networks." Proceedings of the 14th ACM International Conference on Web Search and Data Mining. 2021.
>
> [3] Ong, Ming Lun, Anthony Li, and Mehul Motani. "Explainable and Actionable Machine Learning Models for Electronic Health Record Data." International Conference on Biomedical Engineering. Cham: Springer International Publishing, 2019.
>
> [4] Tsiakmaki, Maria, and Omiros Ragos. "A case study of interpretable counterfactual explanations for the task of predicting student academic performance." 2021 25th International Conference on Circuits, Systems, Communications and Computers (CSCC). IEEE, 2021.
>
> [5] Wijekoon, Anjana, et al. "Counterfactual Explanations for Student Outcome Prediction with Moodle Footprints." SICSA XAI. 2021.
>
> [6] Davis, Randall, et al. "Explainable machine learning models of consumer credit risk." Available at SSRN 4006840 (2022).
>
> [7] Ramaswami, Gomathy, Teo Susnjak, and Anuradha Mathrani. "Supporting students’ academic performance using explainable machine learning with automated prescriptive analytics." Big Data and Cognitive Computing 6.4 (2022): 105.
>
> [8] Smith, Bevan I., Charles Chimedza, and Jacoba H. Bührmann. "Individualized help for at-risk students using model-agnostic and counterfactual explanations." Education and Information Technologies 27.2 (2022): 1539-1558.
>
> [9] Wang, Zijie J., et al. "Gam coach: Towards interactive and user-centered algorithmic recourse." Proceedings of the 2023 CHI Conference on Human Factors in Computing Systems. 2023.
>
> [9] Lacerda, Anisio, et al. "Algorithmic Recourse in Mental Healthcare." 2023 International Joint Conference on Neural Networks (IJCNN). IEEE, 2023.
>
> [10] Hardt, Mendler-Duenner. Performative Prediction: Past and Future. Arxiv. 2024.
>
> [11] MacKenzie. An Engine Not a Camera: How Financial Models Shape Markets. Oxford University Press. 2009.
>
> ### Q3/W2: Assumptions
>
> *Assumption 4.6 seems to be strong; can it be relaxed? Do we have equivalence with this assumption in theorem 4.7. How can we use it in practice?*
>
> Although it is a strong assumption, 4.6 is practically useful since it covers commonly occurring DGPs, such as linear Gaussian data. We prove that the assumption is sufficient for performative validity (Theorem 4.7). It is not necessary for performative validity, as our experiments suggest. Relaxing 4.6 is an interesting avenue of further research.
>
> *How can we use 4.6 in practice?*
>
> For example, if we know that the data is linear Gaussian and thus that Assumption 4.6 is met, we know that ICR is performatively valid.
>
> *What would be the difficulties if Assumptions 4.4 and 4.6 do not apply?*
>
> If Assumptions 4.4 and 4.6 do not apply (and we cannot guarantee that only causal variables influence model and recourse generation), we cannot guarantee performative validity, regardless of which specific recourse method is used. We demonstrate this in Example 4.3. As such, we reveal a previously unknown fundamental limitation of any recourse method!
>
> *Do we need to know the causal graphs in advance to apply the results in the paper?*
>
> Recourse methods, such as CR and ICR, require knowledge of at least the causal graph. Causal knowledge is an inevitable requirement for recourse, since it is an inherently causal problem.
>
> In contrast, for our main results (4.1 & 4.2), you only need to know whether causal sufficiency holds and whether non-causal variables are involved, which is a weaker requirement than knowing the full causal graph. Still, 4.4 and 4.6 can be best assessed given the complete SCM. Extending our work to settings with more limited causal knowledge is an interesting avenue for future work.
>
> ### W1: What is the impact of refitting the model? How does alpha play into the results?
>
> *The impact of retraining the model is not discussed on experiments or theoretically.*
>
> Our results concern the impact of retraining a model on a mixture of pre- and post-recourse data. For instance, Theorem 4.2 shows that if no noncausal variables are involved, the optimal refit is equivalent to the optimal pre-recourse model. Furthermore, all experiments investigate how the conditional distribution and acceptance rates change when the model is refitted.
>
> *In the paper, it is not clear what is the real impact, and it seems very useful to link the mixture model (1) (and alpha), with the number of rejected recourses or previously accepted observations. Could you clarify? What is the appropriate concept to anticipate the shift / amplitude of the drop in performance of the recourses.*
>
> Our theoretical results are primarily concerned with whether a shift occurs, rather than its magnitude, and thus they are independent of the choice of $\alpha$. In Propositions E.1 and F.1 we provide formulas that describe how the distribution shifts when interventions on effects are performed but to predict the strength of the shift in terms of acceptance rates, one would need to make very concrete assumptions about the distribution and decision thresholds. Intuitively, the strength of the shift depends on how strongly the predictor and recourse method rely on non-causal effect variables.
>
> ### Q2: Mathematical proof for the invalidity of CE/CR
> *Can we show mathematically that CE and CRa are not valid ?*
>
> Whether CE and CR are performatively valid depends on the concrete setting. In Theorem 4.2, we prove mathematically that any recourse method, including CE and CR, is performatively valid if no effect variables are involved. In Examples 4.3 and 4.8, we prove mathematically that CE and CR may be performatively invalid in other cases. We commit to update the text to clarify that the Examples concern both CE and CR.
>
> **We hope our clarifications were helpful in addressing your questions. If they were, we’d greatly appreciate it if you would consider updating your score.**

---

### Note · Authors · 2025-08-13

We thank the reviewers for their positive support: The reviewers find the paper to be "well-written", "interesting" and "novel", the story "theoretically and empirically compelling", the approach "likely to inspire follow-up research in this emerging area", and the insights "useful to the community". Throughout the rebuttal and discussion period, we provided a credit scoring example, intuitive explanations of the assumptions, details on the examples, and additional experiments—all of which will be incorporated into the final draft. Thereby, we were able to incorporate the reviewers' feedback and address all concerns.

---

### Decision · Program_Chairs · 2025-09-17

**Decision:**

Accept (poster)

**Comment:**

The paper investigates the performative effects of algorithmic recourse, where recommendations can become invalid after a model is retrained on data from users who have followed them. To address this, the authors introduce performative validity, a novel criterion that uses a causal framework to formally characterize the conditions under which recourse remains effective. Theoretical analysis and experiments on Structural Causal Models are used to evaluate existing methods, demonstrating that while Counterfactual Explanations and Causal Recourse can become invalid, Improvement-Focused Causal Recourse is shown to maintain its validity. The reviewers’ concerns were satisfactorily addressed by the authors during the rebuttal period. Both the reviewers and I found the paper to have high potential impact for the machine learning community.